# BioT5: Enriching Cross-modal Integration in Biology with Chemical Knowledge and Natural Language Associations

**Qizhi Pei[1,5], Wei Zhang[2], Jinhua Zhu[2], Kehan Wu[2], Kaiyuan Gao[3],**
**Lijun Wu[4]\*, Yingce Xia[4]\*, Rui Yan[1,6]\***

[1]Gaoling School of Artificial Intelligence, Renmin University of China
[2]University of Science and Technology of China
[3]Huazhong University of Science and Technology    [4]Microsoft Research
[5]Engineering Research Center of Next-Generation Intelligent Search
and Recommendation, Ministry of Education
[6]Beijing Key Laboratory of Big Data Management and Analysis Methods
{qizhipei,ruiyan}@ruc.edu.cn
{weizhang_cs,teslazhu,wu_2018}@mail.ustc.edu.cn
im_kai@hust.edu.cn   {lijuwu,yinxia}@microsoft.com

## Abstract

Recent advancements in biological research leverage the integration of molecules, proteins, and natural language to enhance drug discovery. However, current models exhibit several limitations, such as the generation of invalid molecular SMILES, underutilization of contextual information, and equal treatment of structured and unstructured knowledge. To address these issues, we propose BioT5, a comprehensive pre-training framework that enriches cross-modal integration in biology with chemical knowledge and natural language associations. BioT5 utilizes SELFIES for 100% robust molecular representations and extracts knowledge from the surrounding context of bio-entities in unstructured biological literature. Furthermore, BioT5 distinguishes between structured and unstructured knowledge, leading to more effective utilization of information. After fine-tuning, BioT5 shows superior performance across a wide range of tasks, demonstrating its strong capability of capturing underlying relations and properties of bio-entities. Our code is available at https://github.com/QizhiPei/BioT5.

## 1 Introduction

Molecules and proteins are two essential bio-entities in drug discovery (Dara et al., 2022). Small molecule drugs have been the cornerstone of the pharmaceutical industry for nearly a century, owing to their unique advantages such as oral availability, diverse modes of action, etc (AstraZeneca, 2023). Proteins serve as the foundation of life science, functioning as drug targets or crucial elements in disease pathways. As illustrated in Figure 1, both

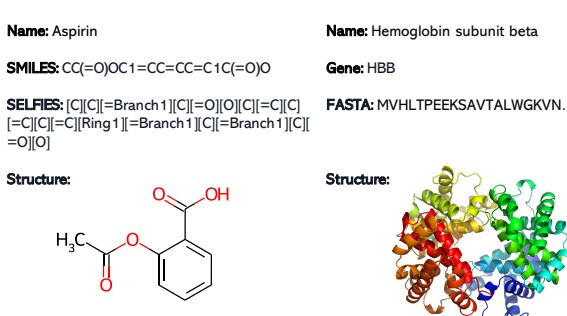

Figure 1: Representations of molecule and protein. Molecule can be represented by its name, bio-sequence (SMILES and SELFIES), and 2D graph structure. Protein can be represented by its name, corresponding gene name, bio-sequence (FASTA), and 3D structure.

molecules and proteins can be represented using sequences. A molecule can be depicted by a SMILES sequence (Weininger, 1988; Weininger et al., 1989), which is derived by traversing the molecular graph through depth-first search and applying specific branching rules. A protein can be represented by a FASTA sequence (Lipman and Pearson, 1985; Pearson and Lipman, 1988), which outlines the amino acids in a protein. The sequential formats of molecules and proteins facilitate the application of Transformer models (Vaswani et al., 2017) and pre-training techniques (Liu et al., 2019; Radford et al., 2019) from natural language processing (NLP) to the biomedical field. Chemberta (Chithrananda et al., 2020) and ESM (Rives et al., 2021; Lin et al., 2022) apply masked language modeling to molecular SMILES and protein FASTA respectively, while MolGPT (Bagal et al., 2022) and ProtGPT2 (Ferruz et al., 2022) leverage GPT-style models for molecular and protein generation.

Scientific literature (Beltagy et al., 2019; Canese and Weis, 2013) and biological databases (Kim

---

\*Corresponding authors: Lijun Wu (lijuwu@microsoft.com), Yingce Xia (yinxia@microsoft.com), and Rui Yan (ruiyan@ruc.edu.cn)

et al., 2023; Boutet et al., 2007) serve as knowledge repositories of molecules and proteins. These resources detail properties, experimental results, and interactions between various bio-entities, which cannot be explicitly inferred from molecular or protein sequences alone. Consequently, a recent trend involves jointly modeling text along with molecules and proteins, allowing the textual descriptions to enhance molecular and protein representations. MolT5 (Edwards et al., 2022) adopts the T5 (Raffel et al., 2020) framework to molecular SMILES and biomedical literature. MolXPT (Liu et al., 2023b) and Galactica (Taylor et al., 2022) are GPT models trained on text and bio-entities, such as SMILES and FASTA sequences. Deep-EIK (Luo et al., 2023) fuses the encoded features from multi-modal inputs using attention (Vaswani et al., 2017) mechanism. Despite their success, there is still significant room for improvement: (i) Prior work often relies on SMILES to represent molecules. However, addressing the issue of generating invalid SMILES remains a challenge to overcome (Edwards et al., 2022; Li et al., 2023). (ii) The contextual information surrounding molecular or protein names could offer valuable insights for understanding the interactions and properties of bio-entities. Developing effective methods to leverage this information merits further attention. (iii) Existing research tends to treat structured data (e.g., molecule-text pairs from databases) and unstructured data (e.g., text sequences in literature) equally. However, structured data could be utilized more effectively to further enhance overall performance.

To address the above challenges, in this paper, we introduce **BioT5**, a comprehensive pre-training framework encompassing text, molecules, and proteins. BioT5 leverages SELFIES (Krenn et al., 2020) to represent small molecules since its advantage over SMILES is that SELFIES offers a more robust and error-tolerant molecular representation, eliminating issues of illegitimate structures often encountered with SMILES. There are mainly two steps for BioT5 pre-training:

(1) *Data collection & processing*: We gather text, molecule, and protein data, as well as existing databases containing molecule-text parallel data and protein-text parallel data. For the text data (PubMed) from the biological domain, we employ named entity recognition and entity linking to extract molecular and protein mentions, replacing them with the corresponding SELFIES or FASTA

sequences. Following Liu et al. (2023b), we refer to such data as "wrapped" text. Text tokens, FASTA sequences, and SELFIES are tokenized independently (see Section 3.2 for more details).

(2) *Model training*: BioT5 utilizes a shared encoder and a shared decoder to process various modalities. The standard T5 employs the "recover masked spans" objective, wherein each masked span and its corresponding part share the same sentinel token. We refer to the aforementioned training objective function as the "T5 objective" for simplicity. There are three types of pre-training tasks: (i) Applying the standard T5 objective to molecule SELFIES, protein FASTA, and general text independently, ensuring that the model possesses capabilities in each modality. (ii) Applying the T5 objective to wrapped text from the biological domain, where all text, FASTA, and SELFIES tokens can be masked and recovered. (iii) For the structured molecule-text data, we introduce a translation objective. Specifically, BioT5 is trained to translate molecule SELFIES to the corresponding description and vice versa. Likewise, the translation objective is applied to protein-text data.

After pre-training, we fine-tune the obtained BioT5 on 15 tasks covering molecule and protein property prediction, drug-target interaction prediction, protein-protein interaction prediction, molecule captioning, and text-based molecule generation. BioT5 achieves state-of-the-art performances on 10 tasks and exhibits results comparable to domain-specific large models on 5 tasks, demonstrating the superior ability of our proposed method. BioT5 model establishes a promising avenue for the integration of chemical knowledge and natural language associations to augment the current understanding of biological systems.

## 2 Related Work

In this section, we briefly review related work about cross-modal models in biology and representations of molecule and protein.

### 2.1 Cross-modal Models in Biology

Language models in the biology field have gained considerable attention. Among these, BioBERT (Lee et al., 2020) and BioGPT (Luo et al., 2022), which are pre-trained on scientific corpora, have been particularly successful in effectively understanding scientific texts. More recently, cross-modal models focusing on jointly modeling

text with bio-sequences have emerged. They can be categorized into the following three groups.

**Cross Text-molecule Modalities** MolT5 (Edwards et al., 2022) is a T5 (Raffel et al., 2020)-based model, which is jointly trained on molecule SMILES and general text corpus. MoSu (Su et al., 2022) is trained on molecular graphs and related textual data using contrastive learning. MolXPT (Liu et al., 2023b) is a GPT (Radford et al., 2018)-based model pre-trained on molecule SMILES, biomedical text, and wrapped text. Different from BioT5, these models all use SMILES to represent molecules, which leads to validity issues when generating molecules.

**Cross Text-protein Modalities** ProteinDT (Liu et al., 2023a) is a multi-modal framework that uses semantically-related text for protein design. BioTranslator (Xu et al., 2023a) is a cross-modal translation system specifically designed for annotating biological instances, such as gene expression vectors, protein networks, and protein sequences, based on user-written text.

**Cross Three or More Biology Modalities** Galactica (Taylor et al., 2022) is a general GPT-based large language model trained on various scientific domains, including scientific paper corpus, knowledge bases (e.g., PubChem (Kim et al., 2023) molecules, UniProt (uni, 2023) protein), codes, and other sources. DeepEIK (Luo et al., 2023) fuses the feature from multi-modal inputs (drugs, proteins, and text). Then attention (Vaswani et al., 2017) mechanism is adopted to do textual information denoising and heterogeneous features integration.

Our work differs from previous studies in several ways: (1) we primarily focus on two biological modalities—molecule, protein-with text serving as a knowledge base and bridge to enrich the underlying relations and properties in the molecule and protein domains; (2) we use multi-task pre-training to model the connections between these three modalities in a more comprehensive manner. (3) we use SELFIES instead of SMILES to represent molecules, which is more robust and resolves the validity issue in molecule generation tasks.

## 2.2 Representations of Molecule and Protein

**Molecule Representation** The representation and modeling of molecules have long been a challenge in bioinformatics. There are many methods to represent a molecule: name, fingerprint (Rogers and Hahn, 2010a), SMILES (Weininger, 1988;

Weininger et al., 1989), InChl (Heller et al., 2013), DeepSMILES (O'Boyle and Dalke, 2018), SELFIES (Krenn et al., 2020), 2D molecular graph, etc. SMILES (Simplified Molecular-Input Line-Entry System), a compact and textual representation of the molecular structure, is the most common method. It employs a sequence of characters to encode atoms, bonds, and other molecular features. However, SMILES has several drawbacks (Krenn et al., 2022), such as the lack of syntactic and semantic robustness, which significantly affects the validity of molecules generated by deep learning models (Edwards et al., 2022). To address this issue, SELFIES (Self-referencing Embedded Strings) is introduced as a 100% robust molecular string representation (Krenn et al., 2020). Every permutation of symbols within the SELFIES alphabet invariably generates a chemically valid molecular structure, ensuring that each SELFIES corresponds to a valid molecule. Unlike existing works introduced in Section 2.1 that use SMILES for molecule representation, we employ SELFIES with separate encoding in BioT5 to achieve 100% validity in downstream molecule generation tasks.

**Protein Representation** Protein can also be represented in various ways, such as by its name, corresponding gene name, FASTA format, or 3D geometric structure. The FASTA format is a common choice for encoding protein sequences, which uses single-letter codes to represent the 20 different amino acids. In BioT5, we also employ FASTA format for protein representation.

Unlike Edwards et al. (2022) and Taylor et al. (2022) that share the dictionary between bio-sequence tokens and nature language tokens, BioT5 uses a separate dictionary and biology-specific tokenization to explicitly distinguish biological modalities. We give further analysis of this in Section 3.2.

## 3 BioT5

The overview of the BioT5 pre-training is illustrated in Figure 2. We combine data from different modalities to perform multi-task pre-training.

### 3.1 Pre-training Corpus

As shown in Figure 2, the pre-training corpus of BioT5 is categorized into three classes: (1) *Single-modal data*, including molecule SELFIES, protein FASTA, and general text. For small molecules, we use the ZINC20 (Irwin et al., 2020) dataset and convert SMILES to SELFIES. For protein

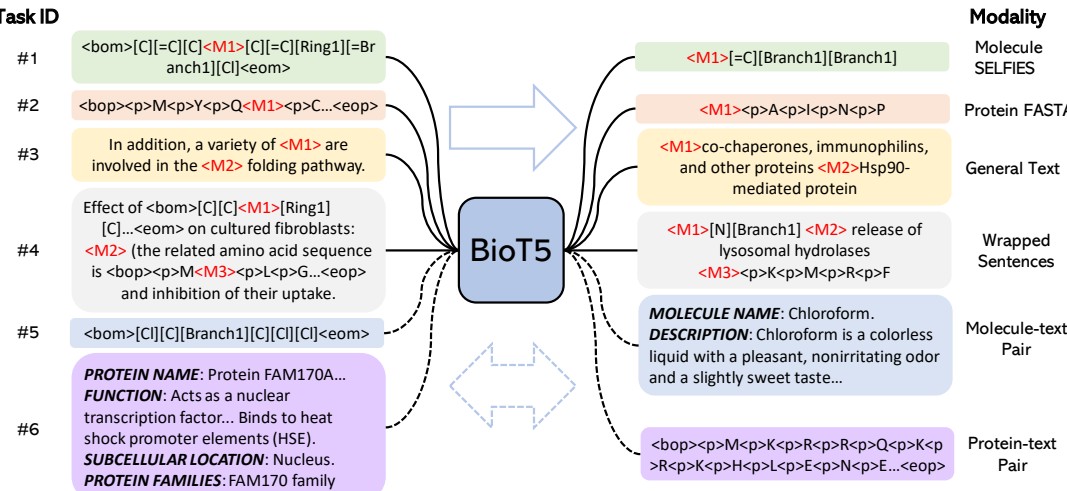

Figure 2: Overview of BioT5 pre-training. The solid line refers to the "T5 objective", which aims to reconstruct the original unmasked input. Each consecutive span of masked tokens is replaced with a sentinel token, depicted as <M1>, <M2>, and <M3>. We apply this objective to molecule SELFIES (task #1), protein FASTA (task #2), general text (task #3), and wrapped text (task #4). The dashed line represents the bidirectional translation between bio-sequences and structured text description (task #5 and #6).

FASTA, we randomly sample proteins from the Uniref50 (Suzek et al., 2007) dataset, filtering out proteins exceeding a specified length, resulting in a collection of 27M proteins For general text, we use the "Colossal Clean Crawled Corpus" (C4) dataset (Raffel et al., 2020). (2) *Wrapped text*, where molecule names are replaced with their corresponding SELFIES and gene names are appended with related protein FASTA. We use 33M PubMed articles (Canese and Weis, 2013) and apply BERN2 (Sung et al., 2022) for named entity recognition. The scientific sentences which are not replaced or appended by bio-sequences are remained as a supplement to general text. The detailed process is depicted in Figure 4 and discussed in Appendix B. (3) *Molecule-description pairs* and *protein-description pairs*. For molecule-text data, we collect 339K molecule SELFIES along with their corresponding names and descriptions from PubChem (Kim et al., 2019), excluding all molecules present in the downstream ChEBI-20 dataset (Edwards et al., 2022) to avoid potential data leakage. For protein-text data, we obtain 569K protein FASTA-description pairs from Swiss-Prot (Boutet et al., 2007), which contains high-quality annotations of various protein properties. Details are left in Appendix E.1.

## 3.2 Separate Tokenization and Embedding

In most previous works, the representation of molecules and proteins has not been modeled with

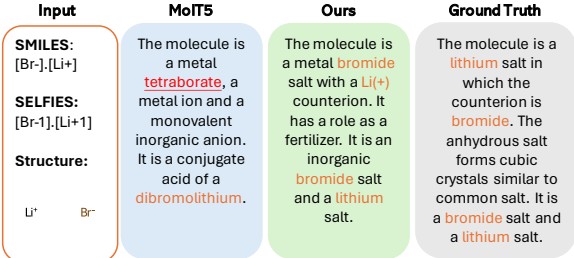

Figure 3: Case for tokenization. MolT5 processes "Br"(bromine atom) as "B" (boron atom) and "r", resulting in incorrect descriptions including tetraborate (related to "B"). BioT5 retains the chemically meaningful group "[Br-1]" as a complete token, thereby producing the correct output.

sufficient attention to detail. MolT5 (Edwards et al., 2022) employs the same dictionary as the original T5, as it starts pre-training from the original T5 checkpoint. The original T5 dictionary is derived from nature language using SentencePiece (Kudo and Richardson, 2018). However, directly utilizing this dictionary for molecule SMILES is suboptimal, as some chemically meaningful tokens, such as functional groups or complete atoms, will be tokenized inaccurately. For example, in the molecule depicted in Figure 3, the bromine atom, symbolized as "Br" in SMILES, is tokenized as "B" (a boron atom) and "r" by MolT5. Consequently, MolT5 incorrectly characterizes this molecule as both dibromolit (related to "Br") and tetraborate (related to "B"). The character-based tokenization of Galac-

tica (Taylor et al., 2022) suffers the same issue.

In addition to the tokenization method, sharing token embeddings for different modalities (Edwards et al., 2022; Taylor et al., 2022) is also questionable. In multilingual tasks, shared embeddings allow models to accurately represent the meanings of borrowed words and cognates, which retain their original meanings across languages. However, molecules, proteins, and text represent entirely distinct languages. The same token within these three different modalities carries different semantic meanings. For example, the token "C" signifies character C in nature language, the carbon atom in molecules, and cysteine (one of the 20 amino acids) in proteins. Studies by Beltagy et al. (2019) and Gu et al. (2021) further emphasize the significance of domain-specific vocabulary.

To address the issues mentioned above, we employ separate vocabularies for molecule, protein, and text. In BioT5, molecule is represented by SELFIES string, where each chemical meaningful atom group is enclosed within brackets and tokenized as a SELFIES token. For example, [C][=C][Br]→[C],[=C],[Br]. For protein, to differentiate amino acids with capital letters in text, we introduce a special prefix <p> for each amino acid. For example, <p>M<p>K<p>R→<p>M,<p>K,<p>R. For text, we use the same dictionary as the original T5. Through this, we explicitly distinguish the semantic space of different modalities, which maintains the inherent integrity of each unique modality and prevents the model from conflating meanings across modalities.

### 3.3 Model and Training

**Model architecture** BioT5 employs the same architecture as T5 models (Raffel et al., 2020). We follow the configuration used in T5-v1.1-base[1]. The vocabulary size of BioT5 is $35,073$, differing from the default configuration as we incorporate separate vocabulary for molecule SELFIES and protein amino acids. In total, the BioT5 model comprises 252M parameters.

**Pre-training** During the pre-training phase, the model is trained in a multi-task way on six tasks that can be classified into three types: (1) Applying T5 objective to each single modality including molecule SELFIES (task #1), protein FASTA (task #2), and general text (task #3) independently. (2)

---

[1] https://huggingface.co/docs/transformers/model_doc/t5v1.1

Applying T5 objective to wrapped text from scientific corpus (task #4). (3) Bidirectional translation for the molecule SELFIES-text pairs (task #5) and protein FASTA-text pairs (task #6). By effectively learning the underlying connections and properties of bio-entities from textual information through these pre-training tasks, BioT5 allows for a holistic understanding of the biological domain, thereby facilitating enhanced prediction and generation abilities in various biological tasks.

**Fine-tuning** BioT5 can be fine-tuned on various downstream tasks involving molecules, proteins, and text. To unify different downstream tasks and reduce the gap between pre-training and fine-tuning (Brown et al., 2020) stage, we adopt the prompt-based fine-tuning (Gao et al., 2021) approach, which facilitates various task formats into a sequence generation format.

## 4 Experiments and Results

We evaluate BioT5 on $15$ well-established downstream tasks, which can be categorized into three types: single-instance prediction, multi-instance prediction, and cross-modal generation. We include details regarding fine-tuning datasets, baselines, and prompts in Appendix F.

For the downstream binary classification tasks presented in Section 4.1 and 4.2, the calculation of evaluation metrics such as AUROC and AUPRC necessitates the soft probability of the predicted label. As we use the prompt-based fine-tuning method, the output is either *Yes* for the positive label or *No* for the negative label. To obtain an appropriate label distribution, following Liu et al. (2023b), we first extract the probabilities of *Yes* and *No* tokens (denoted as $p_{pos}$ and $p_{neg}$ respectively) and normalize them. The resulting probability for positive label is $\frac{p_{pos}}{p_{pos}+p_{neg}}$ and negative label is $\frac{p_{neg}}{p_{pos}+p_{neg}}$.

### 4.1 Single-instance Prediction

#### 4.1.1 Molecule Property Prediction

Molecule property prediction aims to determine whether a given molecule exhibits specific properties. MoleculeNet (Wu et al., 2018) is a widely used benchmark for molecule property prediction, encompassing diverse datasets that cover numerous molecular aspects, such as quantum mechanics, physical chemistry, biophysics, etc. In line with Liu et al. (2023b), we conduct experiments on six binary classification tasks, including BBBP, Tox21, ClinTox, HIV, BACE, and SIDER. Following (Fang

| Dataset | BBBP | Tox21 | ClinTox | HIV | BACE | SIDER | Avg |
|---------|------|-------|---------|-----|------|-------|-----|
| **#Molecules** | 2039 | 7831 | 1478 | 41127 | 1513 | 1427 | - |
| **#Tasks** | 1 | 12 | 2 | 1 | 1 | 27 | - |
| G-Contextual | 70.3±1.6 | 75.2±0.3 | 59.9±8.2 | 75.9±0.9 | 79.2±0.3 | 58.4±0.6 | 69.8 |
| G-Motif | 66.4±3.4 | 73.2±0.8 | 77.8±2.0 | 73.8±1.4 | 73.4±4.0 | 60.6±1.1 | 70.9 |
| GROVER$_{base}$ | 70.0±0.1 | 74.3±0.1 | 81.2±3.0 | 62.5±0.9 | 82.6±0.7 | 64.8±0.6 | 72.6 |
| GROVER$_{large}$ | 69.5±0.1 | 73.5±0.1 | 76.2±3.7 | 68.2±1.1 | 81.0±1.4 | 65.4±0.1 | 72.3 |
| GraphMVP | 72.4±1.6 | 75.9±0.5 | 79.1±2.8 | 77.0±1.2 | 81.2±0.9 | 63.9±1.2 | 74.9 |
| MGSSL | 70.5±1.1 | 76.5±0.3 | 80.7±2.1 | 79.5±1.1 | 79.7±0.8 | 61.8±0.8 | 74.8 |
| MolCLR | 72.2±2.1 | 75.0±0.2 | 91.2±3.5 | 78.1±0.5 | 82.4±0.9 | 58.9±1.4 | 76.3 |
| GEM | 72.4±0.4 | **78.1±0.1** | 90.1±1.3 | 80.6 ± 0.9 | 85.6±1.1 | 67.2±0.4 | 79.0 |
| KV-PLM | 74.6±0.9 | 72.7±0.6 | – | 74.0±1.2 | – | 61.5±1.5 | – |
| Galactica | 66.1 | 68.9 | 82.6 | 74.5 | 61.7 | 63.2 | 69.5 |
| MoMu | 70.5±2.0 | 75.6±0.3 | 79.9±4.1 | 76.2±0.9 | 77.1±1.4 | 60.5±0.9 | 73.3 |
| MolXPT | **80.0 ± 0.5** | 77.1±0.2 | 95.3 ± 0.2 | 78.1±0.4 | 88.4 ± 1.0 | 71.7 ± 0.2 | 81.9 |
| BioT5 | 77.7±0.6 | 77.9±0.2 | **95.4±0.5** | **81.0±0.1** | **89.4±0.3** | **73.2±0.2** | **82.4** |

Table 1: Performance comparison on MoleculeNet (**Best**, Second Best). The evaluation metric is AUROC. The baseline results are mainly sourced from MolXPT (Liu et al., 2023b).

| Model | #Params. | Solubility | Localization |
|-------|----------|-----------|--------------|
| DDE | 205.3K | 59.77 ± 1.21 | 77.43 ± 0.42 |
| Moran | 123.4K | 57.73 ± 1.33 | 55.63 ± 0.85 |
| LSTM | 26.7M | 70.18 ± 0.63 | 88.11 ± 0.14 |
| Transformer | 21.3M | 70.12 ± 0.31 | 75.74 ± 0.74 |
| CNN | 5.4M | 64.43 ± 0.25 | 82.67 ± 0.32 |
| ResNet | 11.0M | 67.33 ± 1.46 | 78.99 ± 4.41 |
| ProtBert | 419.9M | 68.15 ± 0.92 | 91.32 ± 0.89 |
| ProtBert* | 419.9M | 59.17 ± 0.21 | 81.54 ± 0.09 |
| ESM-1b | 652.4M | 70.23 ± 0.75 | **92.40 ± 0.35** |
| ESM-1b* | 652.4M | 67.02 ± 0.40 | 91.61 ± 0.10 |
| BioT5 | 252.1M | **74.65 ± 0.49** | 91.69 ± 0.05 |

Table 2: Performance comparison of different methods on solubility and localization prediction tasks (**Best**, Second Best). The evaluation metric is accuracy. * represents only tuning the prediction head. The baseline results are sourced from PEER (Xu et al., 2022).

et al., 2022), we adopt the scaffold splitting, which is more challenging compared to random splitting.

**Baselines** We compare BioT5 with two types of baselines: (1) pre-trained Graph Neural Network (GNN) using molecular graph as input, which are G-Contextual (Rong et al., 2020), G-Motif (Rong et al., 2020), GROVER$_{base}$ (Rong et al., 2020), GROVER$_{large}$ (Rong et al., 2020), GraphMVP (Liu et al., 2022), MGSSL (Zhang et al., 2021) Mol-CLR (Wang et al., 2022) and GEM (Fang et al., 2022); (2) pre-trained language model baselines, which are KV-PLM (Zeng et al., 2022), Galactica (Taylor et al., 2022), MoMu (Su et al., 2022) and MolXPT (Liu et al., 2023b).

**Results** The results are presented in Table 1 with all statistics derived from three random runs. From

these results, we can see that BioT5 surpasses baselines on most downstream tasks in MoleculeNet. BioT5 exhibits superior performance compared to GNN baselines that are pre-trained on 2D/3D molecular data, underscoring the effectiveness of knowledge in text. Furthermore, BioT5 outperforms other language model baselines, which may be attributed to the presence of molecule property descriptions in scientific contextual text or existing biological database entries.

### 4.1.2 Protein Property Prediction

Protein property prediction is crucial as it provides critical insights into the behavior and functions of proteins. We concentrate on two protein property prediction tasks on PEER benchmark (Xu et al., 2022): protein solubility prediction, which aims to predict whether the given protein is soluble, and protein localization prediction, which is to classify proteins as either "membrane-bound" or "soluble".

**Baselines** We compare BioT5 with three types of baselines provided in PEER benchmark: (1) feature engineers, including two protein sequence feature descriptors: Dipeptide Deviation from Expected Mean (DDE) (Saravanan and Gautham, 2015) and Moran correlation (Moran) (Feng and Zhang, 2000); (2) protein sequence encoders, including LSTM (Hochreiter and Schmidhuber, 1997), Transformers (Vaswani et al., 2017), CNN (O'Shea and Nash, 2015) and ResNet (He et al., 2016); (3) pre-trained protein language models, which are pre-trained using extensive collections of protein FASTA sequences, including ProtBert (Elnaggar et al., 2021) and ESM-1b (Rives et al., 2021). Both

| Method | BioSNAP | | | Human | | BindingDB | | |
|---|---|---|---|---|---|---|---|---|
| | AUROC | AUPRC | Accuracy | AUROC | AUPRC | AUROC | AUPRC | Accuracy |
| SVM | 0.862±0.007 | 0.864±0.004 | 0.777±0.011 | 0.940±0.006 | 0.920±0.009 | 0.939±0.001 | 0.928±0.002 | 0.825±0.004 |
| RF | 0.860±0.005 | 0.886±0.005 | 0.804±0.005 | 0.952±0.011 | 0.953±0.010 | 0.942±0.011 | 0.921±0.016 | 0.880±0.012 |
| DeepConv-DTI | 0.886±0.006 | 0.890±0.006 | 0.805±0.009 | 0.980±0.002 | 0.981±0.002 | 0.945±0.002 | 0.925±0.005 | 0.882±0.007 |
| GraphDTA | 0.887±0.008 | 0.890±0.007 | 0.800±0.007 | 0.981±0.001 | 0.982±0.002 | 0.951±0.002 | 0.934±0.002 | 0.888±0.005 |
| MolTrans | 0.895±0.004 | 0.897±0.005 | 0.825±0.010 | 0.980±0.002 | 0.978±0.003 | 0.952±0.002 | 0.936±0.001 | 0.887±0.006 |
| DrugBAN | 0.903±0.005 | 0.902±0.004 | 0.834±0.008 | 0.982±0.002 | 0.980±0.003 | 0.960±0.001 | 0.948±0.002 | 0.904±0.004 |
| BioT5 | **0.937±0.001** | **0.937±0.004** | **0.874±0.001** | **0.989±0.001** | **0.985±0.002** | **0.963±0.001** | **0.952±0.001** | **0.907±0.003** |

Table 3: Performance comparison on the BindingDB, Human and BioSNAP datasets. (**Best**, Second Best). The baseline results derive from DrugBAN (Bai et al., 2023).

| Model | #Params. | Yeast | Human |
|---|---|---|---|
| DDE | 205.3K | 55.83 ± 3.13 | 62.77 ± 2.30 |
| Moran | 123.4K | 53.00 ± 0.50 | 54.67 ± 4.43 |
| LSTM | 26.7M | 53.62 ± 2.72 | 63.75 ± 5.12 |
| Transformer | 21.3M | 54.12 ± 1.27 | 59.58 ± 2.09 |
| CNN | 5.4M | 55.07 ± 0.02 | 62.60 ± 1.67 |
| ResNet | 11.0M | 48.91 ± 1.78 | 68.61 ± 3.78 |
| ProtBert | 419.9M | 63.72 ± 2.80 | 77.32 ± 1.10 |
| ProtBert* | 419.9M | 53.87 ± 0.38 | 83.61 ± 1.34 |
| ESM-1b | 652.4M | 57.00 ± 6.38 | 78.17 ± 2.91 |
| ESM-1b* | 652.4M | **66.07 ± 0.58** | **88.06 ± 0.24** |
| BioT5 | 252.1M | 64.89 ± 0.43 | 86.22 ± 0.53 |

Table 4: Performance comparison on Yeast and Human datasets (**Best**, Second Best). The evaluation metric is accuracy. * represents only tuning the prediction head. The baseline results derive from PEER (Xu et al., 2022).

ProtBert and ESM-1b are studied with two settings (i) freezing the protein language model parameters and only training the prediction head; (ii) fine-tuning all model parameters.

**Results** The results are displayed in Table 2, with all statistics derived from three random runs. In the protein solubility prediction task, BioT5 outperforms all baselines in PEER (Xu et al., 2022) benchmark. In the protein localization prediction task, BioT5 is the second best among all methods. Notably, ProtBert and ESM-1b are both pre-trained on a large corpus of protein sequences, which is comparable to or even larger than ours. Moreover, these models are two to three times larger than BioT5. These demonstrate the potential of BioT5 for enhanced predictive capabilities in protein property prediction by integrating textual information.

### 4.2 Multi-instance Prediction

#### 4.2.1 Drug-target Interaction Prediction

Drug-target interaction (DTI) prediction plays a crucial role in drug discovery, as it aims to predict whether a given drug (molecule) and target

(protein) can interact with each other. We select three widely-used DTI datasets with a binary classification setting, which are BioSNAP (Zitnik et al., 2018), BindingDB (Liu et al., 2007) and Human (Liu et al., 2015; Chen et al., 2020).

**Baselines** We compare BioT5 with two types of baselines: (1) traditional machine learning methods including SVM (Cortes and Vapnik, 1995) and Random Forest (RF) (Ho, 1995); (2) deep learning methods including DeepConv-DTI (Lee et al., 2019), GraphDTA (Nguyen et al., 2021), MolTrans (Huang et al., 2021) and DrugBAN (Bai et al., 2023), in which drug and target feature are firstly extracted by well-design drug encoder and protein encoder then fused for prediction.

**Results** The results on BioSNAP, Human, and BindingDB datasets are presented in Table 3. All statistics are obtained from five random runs. On BioSNAP and BindingDB datasets, BioT5 consistently outperforms other methods in various performance metrics, including AUROC, AUPRC, and accuracy. For the Human dataset, although deep learning-based models generally exhibit strong performance, the BioT5 model demonstrates a slight advantage over the baseline models. It is worth noting that, in contrast to most deep learning-based baselines, our BioT5 does not rely on a specific design tailored for molecules or proteins. A possible explanation for the superior performance of BioT5 is that the SELFIES and FASTA representations effectively capture the intricate structure and function of molecules and proteins, and the interaction information between them may be well-described in the contextual scientific literature or corresponding text entries in databases.

#### 4.2.2 Protein-protein Interaction Prediction

Protein-protein interaction (PPI) prediction plays a vital role in understanding protein functions and structures, as it aims to determine the potential

| Model | #Params. | BLEU-2 | BLEU-4 | ROUGE-1 | ROUGE-2 | ROUGE-L | METEOR | Text2Mol |
|---|---|---|---|---|---|---|---|---|
| RNN | 56M | 0.251 | 0.176 | 0.450 | 0.278 | 0.394 | 0.363 | 0.426 |
| Transformer | 76M | 0.061 | 0.027 | 0.204 | 0.087 | 0.186 | 0.114 | 0.057 |
| T5-small | 77M | 0.501 | 0.415 | 0.602 | 0.446 | 0.545 | 0.532 | 0.526 |
| T5-base | 248M | 0.511 | 0.423 | 0.607 | 0.451 | 0.550 | 0.539 | 0.523 |
| T5-large | 783M | 0.558 | 0.467 | 0.630 | 0.478 | 0.569 | 0.586 | 0.563 |
| T5-small | 77M | 0.501 | 0.415 | 0.602 | 0.446 | 0.545 | 0.532 | 0.526 |
| MolT5-small | 77M | 0.519 | 0.436 | 0.620 | 0.469 | 0.563 | 0.551 | 0.540 |
| T5-base | 248M | 0.511 | 0.423 | 0.607 | 0.451 | 0.550 | 0.539 | 0.523 |
| MolT5-base | 248M | 0.540 | 0.457 | 0.634 | 0.485 | 0.578 | 0.569 | 0.547 |
| T5-large | 783M | 0.558 | 0.467 | 0.630 | 0.478 | 0.569 | 0.586 | 0.563 |
| MolT5-large | 783M | 0.594 | 0.508 | 0.654 | 0.510 | 0.594 | 0.614 | 0.582 |
| GPT-3.5-turbo (zero-shot) | >175B | 0.103 | 0.050 | 0.261 | 0.088 | 0.204 | 0.161 | 0.352 |
| GPT-3.5-turbo (10-shot MolReGPT) | >175B | 0.565 | 0.482 | 0.623 | 0.450 | 0.543 | 0.585 | 0.560 |
| MolXPT | 350M | 0.594 | 0.505 | 0.660 | 0.511 | 0.597 | 0.626 | 0.594 |
| BioT5 | 252M | **0.635** | **0.556** | **0.692** | **0.559** | **0.633** | **0.656** | **0.603** |

Table 5: Performance comparison on molecule captioning task (**Best**, Second Best). Rouge scores are F1 values. The Text2Mol score between ground truth molecule and corresponding text description is 0.609. The baseline results derive from MolT5 (Edwards et al., 2022), MolXPT (Liu et al., 2023b), and MolReGPT (Li et al., 2023).

| Model | #Params. | BLEU↑ | Exact↑ | Levenshtein↓ | MACCS FTS↑ | RDK FTS↑ | Morgan FTS↑ | FCD↓ | Text2Mol↑ | Validity↑ |
|---|---|---|---|---|---|---|---|---|---|---|
| RNN | 56M | 0.652 | 0.005 | 38.09 | 0.591 | 0.400 | 0.362 | 4.55 | 0.409 | 0.542 |
| Transformer | 76M | 0.499 | 0.000 | 57.66 | 0.480 | 0.320 | 0.217 | 11.32 | 0.277 | 0.906 |
| T5-small | 77M | 0.741 | 0.064 | 27.703 | 0.704 | 0.578 | 0.525 | 2.89 | 0.479 | 0.608 |
| T5-base | 248M | 0.762 | 0.069 | 24.950 | 0.731 | 0.605 | 0.545 | 2.48 | 0.499 | 0.660 |
| T5-large | 783M | 0.854 | 0.279 | 16.721 | 0.823 | 0.731 | 0.670 | 1.22 | 0.552 | 0.902 |
| T5-small | 77M | 0.741 | 0.064 | 27.703 | 0.704 | 0.578 | 0.525 | 2.89 | 0.479 | 0.608 |
| MolT5-small | 77M | 0.755 | 0.079 | 25.988 | 0.703 | 0.568 | 0.517 | 2.49 | 0.482 | 0.721 |
| T5-base | 248M | 0.762 | 0.069 | 24.950 | 0.731 | 0.605 | 0.545 | 2.48 | 0.499 | 0.660 |
| MolT5-base | 248M | 0.769 | 0.081 | 24.458 | 0.721 | 0.588 | 0.529 | 2.18 | 0.496 | 0.772 |
| T5-large | 783M | 0.854 | 0.279 | 16.721 | 0.823 | 0.731 | 0.670 | 1.22 | 0.552 | 0.902 |
| MolT5-large | 783M | 0.854 | 0.311 | 16.071 | 0.834 | 0.746 | 0.684 | 1.20 | 0.554 | 0.905 |
| GPT-3.5-turbo (zero-shot) | >175B | 0.489 | 0.019 | 52.13 | 0.705 | 0.462 | 0.367 | 2.05 | 0.479 | 0.802 |
| GPT-3.5-turbo (10-shot MolReGPT) | >175B | 0.790 | 0.139 | 24.91 | 0.847 | 0.708 | 0.624 | 0.57 | 0.571 | 0.887 |
| MolXPT | 350M | - | 0.215 | - | 0.859 | 0.757 | 0.667 | 0.45 | **0.578** | 0.983 |
| BioT5 | 252M | **0.867** | **0.413** | **15.097** | **0.886** | **0.801** | **0.734** | **0.43** | 0.576 | **1.000** |

Table 6: Performance comparison on text-based molecule generation task (**Best**, Second Best). Following Edwards et al. (2022), BLEU, Exact, Levenshtein, and Validity are computed on all generated molecues while other metrics are computed only on syntactically valid molecules. The Text2Mol score for ground truth is 0.609. The baseline results derive from MolT5 (Edwards et al., 2022), MolXPT (Liu et al., 2023b), and MolReGPT (Li et al., 2023).

interactions between pairs of proteins. Following PEER (Xu et al., 2022) benchmark, we perform fine-tuning on two PPI datasets: Yeast (Guo et al., 2008) and Human (Pan et al., 2010).

**Baselines** The baselines for comparison are the same as that in Section 4.1.2.

**Results** The results are shown in Table 4. All statistics are over three random runs. On two PPI datasets, BioT5 shows superior performance compared to almost all baseline models. Remarkably, BioT5 outperforms both ProtBert and ESM-1b (with full parameters fine-tuned). This result strongly highlights the crucial role of incorporating textual information during the pre-training of BioT5, which effectively establishes profound connections between proteins. Our model, despite being smaller, is able to harness the unstructured

information embedded in scientific text and structured information from biological databases, encapsulating the comprehensive knowledge of proteins in their varying contexts.

### 4.3 Cross-modal Generation

In this section, we evaluate the performance of BioT5 on the cross-modal generation task. Specifically, we fine-tune BioT5 on molecule captioning and text-based molecule generation tasks. These two tasks are proposed by MolT5 (Edwards et al., 2022) and both use the ChEBI-20 dataset (Edwards et al., 2021). The evaluation metrics and some interesting cases are introduced in Appendix D and G.

#### 4.3.1 Molecule Captioning

For the given molecule, the goal of molecule captioning task is to provide a description of the given

molecule. As we use SELFIES sequences to represent molecules, this task can be formulated as an exotic sequence-to-sequence translation task.

**Baselines** The baselines include: RNN (Medsker and Jain, 2001), Transformer (Vaswani et al., 2017), T5 (Raffel et al., 2020), MolT5 (Edwards et al., 2022), GPT-3.5-turbo[2] with zero-shot and 10-shot MolReGPT (Li et al., 2023) settings, and MolXPT (Liu et al., 2023b).

**Results** The results are shown in Table 5. BioT5 only has nearly the same number of parameters as MolT5-base, but outperforms all baseline models in all metrics, including those that have more parameters. The Text2Mol score is 0.603, which is very close to the Text2Mol score of 0.609 between the ground truth molecule and the corresponding description. We can attribute this superior performance to the unstructured contextual knowledge and structured database knowledge induced in BioT5 pre-training, which helps the model learn the intricate relationship between text and molecules.

### 4.3.2 Text-Based Molecule Generation

This is a reverse task of molecule captioning. Given the nature language description of the intended molecule, the goal is to generate the molecule that fits the description.

**Baselines** The compared baselines are the same as baselines in Section 4.3.1.

**Results** The results are presented in Table 6. BioT5 only uses parameters similar to MolT5-base yet delivers superior performance across nearly all metrics. Notably, the exact match score of BioT5 surpasses the MolT5-Large by 32.8% while maintaining a validity of 1.0. This indicates that BioT5 not only generates more relevant molecules corresponding to the given text descriptions, but also ensures a 100% validity for the generated molecules. The overall enhanced performance of BioT5 can be attributed to the incorporation of both contextual and database knowledge, as well as the utilization of SELFIES for molecular representation.

## 5 Conclusions and Future Work

In this paper, we propose BioT5, a comprehensive pre-training framework capable of capturing the underlying relations and properties of bio-entities by leveraging both structured and unstructured data sources with 100% robust molecular representation. Our method effectively enriches cross-modal integration in biology with chemical knowledge and natural language associations, demonstrating notable improvements in various tasks.

For future work, we aim to further enrich our model by incorporating additional biological data types, such as genomics or transcriptomics data, to create a more holistic biological pre-training framework. Additionally, we plan to evaluate the interpretability of BioT5 predictions, aiming to provide more insights into the biological systems under study. Thus, we foresee our work sparking further innovation in the use of AI models in the field of computational biology, ultimately leading to a deeper understanding of biological systems and facilitating more efficient drug discovery.

## 6 Limitations

One limitation of BioT5 is conducting full-parameter fine-tuning on each downstream task. This is done because we do not observe generalization ability among different downstream tasks using instruction-tuning (Wei et al., 2022) method. Another reason is that combining data from different tasks using instructions results in data leakage. For example, have noticed overlaps between the training set of BindingDB and the test sets of BioSNAP and Human. Additionally, we only demonstrate the ability of BioT5 in text, molecule, and protein modalities. Numerous other biological modalities, such as DNA/RNA sequences and cells, exist, and there are many other tasks within a single modality or across multiple modalities. Moreover, BioT5 primarily focuses on the sequence format of bio-entities, yet other formats, such as 2D or 3D structures, also hold significant importance. We leave further exploration of these to future work.

## 7 Acknowledgements

This work was supported by the National Key Research and Development Program of China (No. 2020YFB1406702), National Natural Science Foundation of China (NSFC Grant No. 62122089), Beijing Outstanding Young Scientist Program NO. BJJWZYJH012019100020098, and Intelligent Social Governance Platform, Major Innovation & Planning Interdisciplinary Platform for the "Double-First Class" Initiative, Renmin University of China, the Fundamental Research Funds for the Central Universities, and the Research Funds of Renmin University of China.

---

[2]https://openai.com/blog/openai-api

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

## A Reproducibility

The codes for our BioT5 are available at `https://github.com/QizhiPei/BioT5`.

## B NER and Entity Linking Process

We follow KV-PLM (Zeng et al., 2022) and MolXPT (Liu et al., 2023b) to conduct Named Entity Recognition (NER) and Entity Linking for the bio-entity names appearing in the scientific text. More specifically, we firstly utilize BERN2 (Sung et al., 2022), an advanced neural Named Entity Recognition (NER) tool in biomedical fields, to identify all instances of molecule or protein mentions. Subsequently, we map them to corresponding entities within publicly accessible knowledge bases. For molecule, we use ChEBI (Hastings et al., 2016) and MeSH (Lipscomb, 2000) database, and for protein we use NCBI Gene (Brister et al., 2015) database. Then we can get the corresponding molecule SELFIES and protein FASTA for the matched entities. As shown in Figure 4, for

Figure 4: Wrapped text matching and mapping process.

molecule, we directly replace all the detected names with its SELFIES string; for protein, due to the length limitation, if a sentence consists of more than one protein entity, we only randomly choose one to append the protein FASTA to the name. The motivation for appending protein FASTA instead of replacing is that the genes are transcribed and translated to generate proteins. Therefore, unlike the molecule names directly representing the molecule, the relation between gene names and protein FASTA is indirect. Note that the replacement or appendage will not happen in every sentence. Only those with detected bio-entities will be done the above process.

## C Dictionary and SELFIES Conversion

For molecule-related datasets, when only SMILES is provided, we utilize *selfies*[3] package to convert SMILES into SELFIES.

## D Molecule-Text Generation Metrics

We follow Edwards et al. (2022) to use the same evaluation metrics for molecule captioning and text-based molecule generation tasks. To ensure a fair comparison, we convert the molecule SEIFLES to SMILES before calculating these metrics.

### D.1 Molecule Captioning Metrics

In the molecule caption task, NLP metrics like BLEU (Papineni et al., 2002), ROUGE (Lin, 2004), and METEOR (Banerjee and Lavie, 2005) are utilized to evaluate the closeness of the generated description to the ground truth description. We also adopt *Text2Mol* metric, which is proposed by Edwards et al. (2021) and employ pre-trained models to measure the similarity between the description and ground truth molecule. Higher similarity means that the given text description is more relevant to the molecule, and the Text2Mol score between the ground truth description and molecule is also computed for comparison.

### D.2 Text-based Molecule Generation Metrics

Since molecules can be represented in bio-sequence structure, NLP metrics like BLEU (Papineni et al., 2002) and Exact Match scores between generated and ground truth SMILES are directly applied for evaluation. Additionally, we also report performance on molecule-specific metrics: three molecule fingerprints (FTS) similarity scores- MACCS (Durant et al., 2002), RDK (Schneider et al., 2015), and Morgan (Rogers and Hahn, 2010a); Levenshtein distance (Miller et al., 2009); FCD score (Preuer et al., 2018), which measures molecule similarities according to biological information based on pre-trained "ChemNet"; validity, which is the percentage of the valid SMILES that can be processed by RDKit (Landrum, 2021). The *Text2Mol* metric is also used to measure the similarity between the molecule SMILES and ground truth description.

---

[3]`https://github.com/aspuru-guzik-group/selfies`

# E Pre-training Details

## E.1 Special Tokens

In the pre-training of BioT5, we conduct translation tasks on molecule-text pairs and protein-text pairs extracted from PubChem (Kim et al., 2023) and Swiss-Prot (Boutet et al., 2007) separately. We format the text description from these database entries using special tokens, which serve as anchors for embedding scientific context and structure. For molecule, we use *MOLECULE NAME* and *DESCRIPTION* to represent its name and description including properties, functions, etc. For protein, similar to Xu et al. (2023b), we use *PROTEIN NAME*, *FUNCTION*, *SUBCELLULAR LOCATION*, and *PROTEIN FAMILIES* to represent its name, functions, location and topology in the cell, and families it belongs to. A complete text description is created by concatenating these fields sequentially, omitting any missing fields. Through special tokens, we can effectively encode the intricate information associated with each bio-entity.

## E.2 Hyper-parameters

We use the codebase *nanoT5* (Nawrot, 2023) for BioT5 pre-training. We pre-train BioT5 for 350K steps on eight NVIDIA 80GB A100 GPUs. The batch size is 96 per GPU, in which a batch includes six types of data. The "translation" directions for molecule-text and protein-text pair are randomly selected for each sample with a probability of 0.5. We use AdamW (Loshchilov and Hutter, 2019) with Root Mean Square (RMS) scaling Optimizer for optimization. The learning rate scheduler is cosine annealing with the base learning rate set to $1e-2$ and the minimum learning rate set to $1e-5$. The number of warm-up steps is 10,000 and the dropout rate is $0.0$. The maximum input length for pre-training is $512$. Unlike absolute position encodings, T5 (Raffel et al., 2020) use relative position encodings. This makes the model flexible to inputs of different lengths, which is helpful for downstream fine-tuning.

# F Fine-tuning Details

In this section, we provide details about downstream tasks, including datasets, compared baselines, and prompts. Some statistics about downstream tasks are shown in Table 7 When displaying prompts, ⟨SELFIES⟩ refers to the molecule SELFIES, and ⟨FASTA⟩ refers to the protein FASTA.

## F.1 Single-instance Prediction

### F.1.1 Molecule Property Prediction

All the datasets are split using an $8:1:1$ ratio for train, validation, and test, respectively. We use the scaffold splitting method, in which molecules are categorized according to the Bemis-Murcko scaffold representation.

**Datasets**

(1) The BBBP (Blood-Brain Barrier Penetration) is curated with the intention of aiding the modeling and forecasting of barrier permeability. It comprises compounds that are categorized using binary labels, indicating whether they can penetrate the blood-brain barrier.

(2) The Tox21 ("Toxicology in the 21st Century") initiative established a publicly accessible database that quantifies the toxicity levels of various compounds. The dataset encompasses qualitative toxicity assessments (binary labels) for approximately 8,000 compounds, targeting 12 distinct biological pathways such as nuclear receptors and stress response mechanisms.

(3) The ClinTox dataset contrasts FDA-approved drugs with those that have been unsuccessful in clinical trials owing to toxicity issues. This dataset incorporates two classification objectives for 1,491 drug compounds with established chemical structures: (i) Presence or absence of toxicity in clinical trials; (ii) approved or unapproved by FDA.

(4) The HIV dataset assesses the inhibitory potential of over 40,000 compounds on HIV replication. The screening outcomes were classified into three categories: Confirmed Inactive (CI), Confirmed Active (CA), and Confirmed Moderately Active (CM). Subsequently, the latter two labels were combined, transforming the task into a binary classification between inactive (CI) and active (CA and CM) categories.

(5) The BACE dataset presents quantitative IC50 values and qualitative binary labels for a collection of inhibitors targeting human beta-secretase 1 (BACE-1).

(6) The SIDER (Side Effect Resource) is a comprehensive database that consists of marketed drugs and their corresponding adverse drug reactions (ADR). The drug side effects in SIDER are organized into 27 system organ classes, adhering to the MedDRA classifications. This dataset encompasses data for 1,427 approved drugs.

**Baselines**

(1) GROVER (Rong et al., 2020) incorporates Mes-

| Task/Dataset | Task Type | #Train | #Validation | #Test |
|---|---|---|---|---|
| **Molecule Property Prediction** | | | | |
| **BBBP** | Molecule-wise Classification | 1,631 | 204 | 204 |
| **Tox21** | Molecule-wise Classification | 6,264 | 783 | 784 |
| **ClinTox** | Molecule-wise Classification | 1,181 | 148 | 148 |
| **HIV** | Molecule-wise Classification | 32,901 | 4,113 | 4,113 |
| **BACE** | Molecule-wise Classification | 1,210 | 151 | 152 |
| **SIDER** | Molecule-wise Classification | 1,141 | 143 | 143 |
| **Protein Property Prediction** | | | | |
| **Solubility prediction** | Protein-wise Classification | 62,478 | 1,999 | 1,999 |
| **Localization prediction** | Protein-wise Classification | 5,184 | 1,749 | 1,749 |
| **Drug-target Interaction Prediction** | | | | |
| **BioSNAP** | Molecule-protein Classification | 19,224 | 2,747 | 5,493 |
| **Human** | Molecule-protein Classification | 4,197 | 600 | 1,200 |
| **BindingDB** | Molecule-protein Classification | 50,149 | 5,604 | 5,505 |
| **Protein-protein Interaction Prediction** | | | | |
| **Yeast** | Protein-pair Classification | 4,945 | 394 | 394 |
| **Human** | Molecule-pair Classification | 35,669 | 237 | 237 |
| **Molecule Captioning and Text-based Molecule Generation** | | | | |
| **ChEBI-20** | Molecule-text Translation | 26,407 | 3,301 | 3,300 |

Table 7: Downstream task descriptions, including task or dataset name, type, and the size of each split.

sage Passing Networks within a Transformer-style architecture and is pre-trained on large-scale molecular dataset without any supervision. G-Contextual and G-Motif are two variants of GROVER, which are pre-trained on contextual property prediction task and motif prediction task, respectively.

(2) GraphMVP (Liu et al., 2022) employs self-supervised learning by capitalizing on the correspondence and consistency between molecule 2D topological structures and 3D geometric views.

(3) MGSSL (Zhang et al., 2021) incorporates a novel self-supervised motif generation framework for Graph Neural Networks.

(4) MolCLR (Wang et al., 2022) is a self-supervised learning framework that capitalizes on substantial unlabelled unique molecules (approximately 10 million)

(5) GEM (Fang et al., 2022) features a specially designed geometry-based graph neural network architecture and several dedicated geometry-level self-supervised learning strategies to capture molecular geometry knowledge effectively.

(6) KV-PLM (Zeng et al., 2022) is a BERT-based model designed for molecular representation learning, in which molecule SMILES are appended after its name during the pre-training process. This combination of molecular names and SMILES sequences allows the model to capture both textual and structural information, thereby enhancing its performance in various downstream tasks.

(7) Galactica (Taylor et al., 2022) is a large GPT-based language model which is pre-trained on various corpus like papers, codes, SMILES, protein sequences, etc.

(8) MoMu (Su et al., 2022) is pre-trained using molecular graphs and their semantically related textual data through contrastive learning.

(9) MolXPT (Liu et al., 2023b) is a unified GPT-based language model for text and molecules pre-trained on "wrapped" text, where molecule names are replaced with corresponding SMILES.

**Prompts**

For the six MoleculeNet datasets mentioned above, the prompts only differ in the Task Definition. Therefore, we will only provide the Instruction and Output for the first dataset, and the remaining datasets will follow the same format.

(1) BBBP

Task Definition: *Definition: Molecule property prediction task (a binary classification task) for the BBBP dataset. The blood-brain barrier penetration (BBBP) dataset is designed for the model-*

*ing and prediction of barrier permeability. If the given molecule can penetrate the blood-brain barrier, indicate via "Yes". Otherwise, response via "No".*

Instruction: *Now complete the following example - Input: Molecule: ⟨bom⟩⟨SELFIES⟩⟨eom⟩ Output:.*

Output: *Yes* for inhibitor and *No* instead.

(2) Tox21

Task Definition: *Definition: Molecule property prediction task (a binary classification task) for the Tox21 dataset. The Tox21 dataset contains qualitative toxicity measurements for 8k compounds on 12 different targets, including nuclear receptors and stress response pathways. If the given molecule can activate/change/affect ⟨target⟩, indicate via "Yes". Otherwise, response via "No".* where ⟨target⟩ represents the corresponding receptor, domain, element, gene, potential, or pathway for each subtask.

(3) ClinTox

Task Definition: *Definition: Molecule property prediction task (a binary classification task) for the ClinTox dataset. The ClinTox dataset compares drugs approved by the FDA and drugs that have failed clinical trials for toxicity reasons. If the given molecule is ⟨Subtask⟩, indicate via "Yes". Otherwise, response via "No".* where the ⟨Subtask⟩ is either *toxic* or *FDA approved*.

(4) HIV

Task Definition: *Definition: Molecule property prediction task (a binary classification task) for the HIV dataset. The HIV dataset was introduced by the Drug Therapeutics Program (DTP) AIDS Antiviral Screen, which tested the ability to inhibit HIV replication for over 40,000 compounds. If the given molecule can inhibit HIV replication, indicate via "Yes". Otherwise, response via "No".*

(5) BACE

Task Definition: *Definition: Molecule property prediction task (a binary classification task) for the BACE dataset. The BACE dataset provides qualitative (binary label) binding results for a set of inhibitors of human beta-secretase 1 (BACE-1). If the given molecule can inhibit BACE-1, indicate via "Yes". Otherwise, response via "No".*

(6) SIDER

Task Definition: *Definition: Molecule property prediction task (a binary classification task) for the SIDER dataset. The Side Effect Resource (SIDER) is a dataset of marketed drugs and adverse drug reactions (ADR). If the given molecule can cause the side effect of ⟨side effect⟩, indicate via* "Yes". *Otherwise, response via "No".* where ⟨side effect⟩ refers to the corresponding side effects for each subtask.

### F.1.2 Protein Property Prediction

**Datasets**

(1) Solubility prediction is to predict whether a protein is soluble or not. We follow the same splitting method with DeepSol (Khurana et al., 2018).

(2) Localization prediction aims predict whether a protein is "membrane-bound" or "soluble", which is a simple version of subcellular localization prediction task. We follow the same splitting method with DeepLoc (Armenteros et al., 2017).

**Baselines**

(1) Feature engineers. The DDE (Dipeptide Deviation from Expected Mean) (Saravanan and Gautham, 2015) feature descriptor, consisting of 400 dimensions, is based on the dipeptide frequency within a protein sequence. The Moran feature descriptor (Moran correlation) (Feng and Zhang, 2000), with 240 dimensions, characterizes the distribution of amino acid properties within a protein sequence.

(2) Protein sequence encoders, including LSTM (Hochreiter and Schmidhuber, 1997), Transformers (Vaswani et al., 2017), CNN (O'Shea and Nash, 2015) and ResNet (He et al., 2016). The amino acid features in the last layer are aggregated for final prediction.

(3) Pre-trained protein language models. ProtBert (Elnaggar et al., 2021) and ESM-1b (Rives et al., 2021) are both pre-trained on a massive dataset of protein sequences using the masked language modeling (MLM) objective. Specifically, ProtBert is pre-trained on $2.1$ billion protein sequences obtained from the BFD database (Steinegger and Söding, 2018), while ESM-1b is pre-trained on a smaller dataset of $24$ million protein sequences sourced from UniRef50 (Suzek et al., 2007).

**Prompts**

(1) Solubility prediction

Task Definition: *Protein solubility prediction task (a binary classification task) for the solubility dataset. If the given protein is soluble, indicate via "Yes". Otherwise, response via "No".*

Instruction *Now complete the following example - Input: Protein: ⟨bom⟩⟨FASTA⟩⟨eom⟩ Output:.*

Output: *Yes* for soluble protein or *No* instead.

(2) Localization prediction

Task Definition: *Protein subcellular localization task (a binary classification task). If the given protein is membrane-bound, indicate via "Yes". Otherwise (the protein is soluble), response via "No".*
Instruction *Now complete the following example - Input: Protein: ⟨bom⟩⟨FASTA⟩⟨eom⟩ Output:.*
Output: *Yes* for membrane-bound protein or *No* for soluble protein.

### F.2 Multi-instance Prediction

#### F.2.1 Drug-target Interaction Prediction

**Datasets**

(1) BioSNAP (Zitnik et al., 2018) is derived from the DrugBank database (Wishart et al., 2018) and was created by Huang et al. (2021) and Zitnik et al. (2018). It consists of 4,510 drugs and 2,181 proteins. This dataset is balanced, containing both validated positive interactions and an equal number of randomly selected negative samples from unseen pairs.

(2) BindingDB (Liu et al., 2007) is an accessible online database that contains experimentally validated binding affinities. Its main focus is on the interactions between small drug-like molecules and proteins. We follow Bai et al. (2023) to use a modified version of the BindingDB dataset, which was previously constructed by Bai et al. (2021) with reduced bias.

(3) Human (Liu et al., 2015; Chen et al., 2020) is constructed with the inclusion of highly credible negative samples. Following Bai et al. (2023), we also use a balanced version of the Human dataset, which contains an equal number of positive and negative samples.

**Baselines**

We compare the performance of BioT5 with the following six models on DTI task.

(1) Support Vector Machine (Cortes and Vapnik, 1995) (SVM) on the concatenated fingerprint ECFP4 (Rogers and Hahn, 2010b) (extended connectivity fingerprint, up to four bonds) and PSC (Cao et al., 2013) (pseudo-amino acid composition) features.

(2) Random Forest (Ho, 1995) (RF) on the concatenated fingerprint ECFP4 and PSC features.

(3) DeepConv-DTI (Lee et al., 2019) uses a fully connected neural network to encode the ECFP4 drug fingerprint and a Convolutional Neural Network (CNN) along with a global max-pooling layer to extract features from protein sequences. Then the drug and protein features are concatenated and

fed into a fully connected neural network for final prediction.

(4) GraphDTA (Nguyen et al., 2021) uses graph neural networks (GNNs) for the encoding of drug molecular graphs, and a CNN is used for the encoding of protein sequences. The derived vectors of the drug and protein representation are concatenated for interaction prediction.

(5) MolTrans (Huang et al., 2021) uses transformer architecture to encode drug and protein. Then a CNN-based interaction module is used to capture their interactions.

(6) DrugBAN (Bai et al., 2023) use Graph Convolution Network (GCN) (Kipf and Welling, 2017) and 1D CNN to encode drug and protein sequences. Then a bilinear attention network are adopted to learn pairwise local interactions between drug and protein. The resulting joint representation is decoded by a fully connected neural network.

**Prompts**

Task Definition: *Definition: Drug target interaction prediction task (a binary classification task) for the ⟨Dataset⟩ dataset. If the given molecule and protein can interact with each other, indicate via "Yes". Otherwise, response via "No".* where ⟨Dataset⟩ is one of the three DTI datasets mentioned above.
Instruction: *Now complete the following example - Input: Molecule: ⟨bom⟩⟨SELFIES⟩⟨eom⟩ Protein: ⟨bom⟩⟨FASTA⟩⟨eom⟩ Output:.*
Output: *Yes* for positive label or *No* instead.

#### F.2.2 Protein-protein Interaction Prediction

**Datasets**

(1) Yeast (Guo et al., 2008) involves determining whether two yeast proteins interact or not. The negative pairs are derived from distinct subcellular locations. Following (Xu et al., 2022), the dataset is split and removed redundancy according to protein sequences similarity, which allows for the evaluation of generalization across dissimilar protein sequences.

(2) Human (Pan et al., 2010) involves determining whether two human proteins interact or not. It comprises positive protein pairs sourced from the Human Protein Reference Database (HPRD) (Peri et al., 2003) and negative pairs derived from different subcellular locations. The dataset splitting scheme is similar to that of Yeast PPI prediction with an $8 : 1 : 1$ ratio for train/validation/test.

**Baselines** The compared baselines are the same as the protein property prediction task in Sec-

tion F.1.2.

**Prompts**

Task Definition: *Protein protein interaction prediction task (a binary classification task) for the* ⟨Dataset⟩ *dataset. If the given two yeast proteins (Protein_A and Protein_B) can interact with each other, indicate via "Yes". Otherwise, response via "No".* where ⟨Dataset⟩ *is either* yeast *or* human.

Instruction: *Now complete the following example - Input: Protein_A:* ⟨bom⟩⟨FASTA⟩⟨eom⟩ *Protein_B:* ⟨bom⟩⟨FASTA⟩⟨eom⟩ *Output:*.

Output: *Yes* for positive label or *No* instead.

### F.3 Cross-modal Generation

#### F.3.1 Molecule Captioning

**Datasets**

We use ChEBI-20 dataset created by Text2mol (Edwards et al., 2021), which consists of $33,010$ molecule-text pairs and 20 means each text description has more than 20 words. The dataset is split into $8:1:1$ for train, validation, and test.

**Baselines**

(1) RNN (Medsker and Jain, 2001) with 4-layer bidirectional encoder is trained from scratch on ChEBI-20 dataset.

(2) Transformer (Vaswani et al., 2017) containing 6 encoder and decoder layers is trained from scratch on ChEBI-20 dataset.

(3) T5 (Raffel et al., 2020) is directly fine-tuned on ChEBI-20 dataset from public checkpoints [4] with three different model sizes: small, base and large. Note that no molecule domain knowledge is introduced in the original T5 pre-training.

(4) MolT5 (Edwards et al., 2022) is jointly trained on molecule SMILES from ZINC-15 dataset (Sterling and Irwin, 2015) and general text from C4 dataset (Raffel et al., 2020) so that MolT5 has prior knowledge about these two domains. It also contains three different sizes: small, base and large. Then they are further fine-tuned on ChEBI-20 dataset.

(5) GPT-3.5-turbo (Li et al., 2023) is used by directly call OpenAI API without further fine-tuning. The input includes five parts as Li et al. (2023): role identification, task description, examples, output instruction, and user input prompt. The examples are retrieved by Morgan Fingerprint (Butina, 1999) similarity for molecule captioning task and

---

[4] https://github.com/google-research/text-to-text-transfer-transformer/blob/main/released_checkpoints.md#t511

by BM25 (Robertson and Zaragoza, 2009) for text-based molecule generation task.

(6) MolXPT (Liu et al., 2023b) is jointly trained on molecule SMILES from PubChem (Kim et al., 2023), biomedical text from PubMed (Canese and Weis, 2013), and "wrapped" text in which molecule names are replaced with corresponding SMILES.

**Prompts**

Different from the classification task in which the ground truth output is either *Yes* or *No*, the output for molecule captioning task is text sequence.

Task Definition: *Definition: You are given a molecule SELFIES. Your job is to generate the molecule description in English that fits the molecule SELFIES.*

Instruction: *Now complete the following example - Input: <bom>⟨SELFIES⟩<eom> Output:*.

Output: ⟨Text Description⟩

#### F.3.2 Text-based molecule generation

This is the reverse task of molecule captioning. The input is the text description of the desired molecule and the output is the corresponding molecule SELFIES. The datasets and compared baselines are the same with molecule captioning in Section F.3.1 so will only provide the prompts here.

**Prompts**

Task Definition: *Definition: You are given a molecule description in English. Your job is to generate the molecule SELFIES that fits the description.*

Instruction: *Now complete the following example - Input: ⟨Text Description⟩ Output:*.

Output: *<bom>*⟨SELFIES⟩*<eom>*

## G Case Study

In this section, we show several example outputs from different models in molecule captioning and text-based molecule generation tasks. Figure 5 shows the cases for the molecule captioning task. In example (1), the description of BioT5 matches the ground truth best, successfully localizing the position of the substituent group and "member of pyridines and an aryl thiol". In example (2), MolT5 mistakenly describes that the molecule contained boron, while BioT5's description is more accurate. In example (3), while MolT5 generates repetitive output, BioT5 and T5 generate semantically coherent output, and BioT5's output matches better with ground truth. For a complex molecule in example (4), the output of BioT5 is more holistic and accurate. Notably, only BioT5 describes this molecule

as an inhibitor of SARS coronavirus main proteinase, which may come from our integration with protein knowledge. Figure 6 shows the cases for the text-based molecule generation task. From the cases, we have several findings: (i) BioT5 is more likely to generate molecules that exactly match the ground truth. (ii) By using SELFIES, BioT5 will not generate invalid molecules, especially for complex and longer molecules shown in examples (3) and (4). (iii) Some molecules are actually short proteins. Example (3) shows a molecule that is a 33-membered polypeptide, which consists of 33 amino acid residues joined in sequence. Therefore, the boundary between proteins and molecules may not always be distinct, and leveraging information from both can provide reciprocal benefits.

**Figure 5: Molecule captioning cases.**

| # | Input | T5 | MolT5 | Ours | Ground Truth |
|---|-------|----|-------|------|--------------|
| 1 | (thiopyridinone structure) | The molecule is a thiol that is thiol substituted by a sulfanyl group at position 4. It has a role as a metabolite. It is a thiol and a member of benzenes. It derives from a hydride of a thiol. | The molecule is a monothiocarbamic ester resulting from the formal condensation of thiocyanic acid with benzene. It is a member of thiocarbamic acids and a monothiocarbamic ester. | The molecule is pyridine substituted at position 2 by a sulfanyl group. It has a role as a corrosion inhibitor and an allergen. It is a member of pyridines and an aryl thiol. | The molecule is pyridine substituted at C-2 by a sulfanyl group. It has a role as a fluorescence quencher and an allergen. It is a member of pyridines and an aryl thiol. |
| 2 | Li⁺   Br⁻ | The molecule is a metal halide, a metal cation and a lithium molecular entity. It has a role as an osmolyte and a flame retardant. | The molecule is a metal tetraborate, a metal ion and a monovalent inorganic anion. It is a conjugate acid of a dibromolithium. | The molecule is a metal bromide salt with a Li(+) counterion. It has a role as a fertilizer. It is an inorganic bromide salt and a lithium salt. | The molecule is a lithium salt in which the counterion is bromide. The anhydrous salt forms cubic crystals similar to common salt. It is a bromide salt and a lithium salt. |
| 3 | (ethyltrimethylammonium iodide structure) | The molecule is a quaternary ammonium salt that is ammonia in which the hydrogens at the 2- and 3-positions are replaced by methyl groups. It has a role as a cardiotoxic agent. It is a quaternary ammonium salt and a quaternary ammonium salt. It contains a tetramethylammonium. | The molecule is the iodo-iodo-iodo-iodo-iodo-iodo-iodo-iodo-iodo-iodo-iodo-iodo-iodo-iodo-iodo-iodo-iodo-iodo-iodo-iodo derivative of an iodo-iodo-iodo-iodo-iodo-iodo-iodo-iodo-iodo-iodo-iodo-iodo-iodo-iodo-iodo-iodododododo[...] | The molecule is a quaternary ammonium salt that is the monoiodide salt of ethyltrimethylammonium. It is a quaternary ammonium salt, an organic iodide salt and a quaternary ammonium salt. It contains an ethyltrimethylammonium. | The molecule is a quaternary ammonium salt whose basic unit comprises an ethyltrimethylammonium cation and an iodide anion. It is a quaternary ammonium salt and an iodide salt. |
| 4 | (peptide structure) | The molecule is a tetrapeptide composed of L-aspartic acid, L-phenylalanine and two L-proline units joined by peptide linkages. It has a role as a metabolite. It derives from a L-aspartic acid, a L-phenylalanine and a L-proline. | The molecule is a pentapeptide consisting of L-codeoxylimonoyl, L-phenylalanine and L-(2-naphthyl)acetamide residues joined in sequence. It has a role as a metabolite. It is a polypeptide, a member of naphthalenes and a pentapeptide. | The molecule is a secondary carboxamide resulting from the formal condensation of the carboxy group of 1H-indole-2-carboxylic acid with the primary amino group of (2S,3R)-cyclohexyl-L-alanine. It is an inhibitor of SARS coronavirus main proteinase and inhibits SARS-CoV-2 replication in cell culture (EC50 = 0.72 muM). It has a role as an EC 3.4.22.69 (SARS coronavirus main proteinase) inhibitor and an anticoronaviral agent. It is a secondary carboxamide, a member of pyrrolidin-2-ones, an oligopeptide, an indolecarboxamide and a L-alanine derivative. | The molecule is a secondary carboxamide resulting from the formal condensation of the carboxy group of 1H-indole-2-carboxylic acid with the primary amino group of 3-cyclohexyl-N-(2S)-1-oxo-3-[(3S)-2-oxopyrrolidin-3-yl]propan-2-yl-L-alaninamide. It is an inhibitor of SARS coronavirus main proteinase and inhibits SARS-CoV-2 replication in cell culture (EC50 = 0.53 muM). It has a role as an EC 3.4.22.69 (SARS coronavirus main proteinase) inhibitor and an anticoronaviral agent. It is an indolecarboxamide, a member of pyrrolidin-2-ones, an aldehyde, a secondary carboxamide and an oligopeptide. |

Figure 5: Molecule captioning cases.

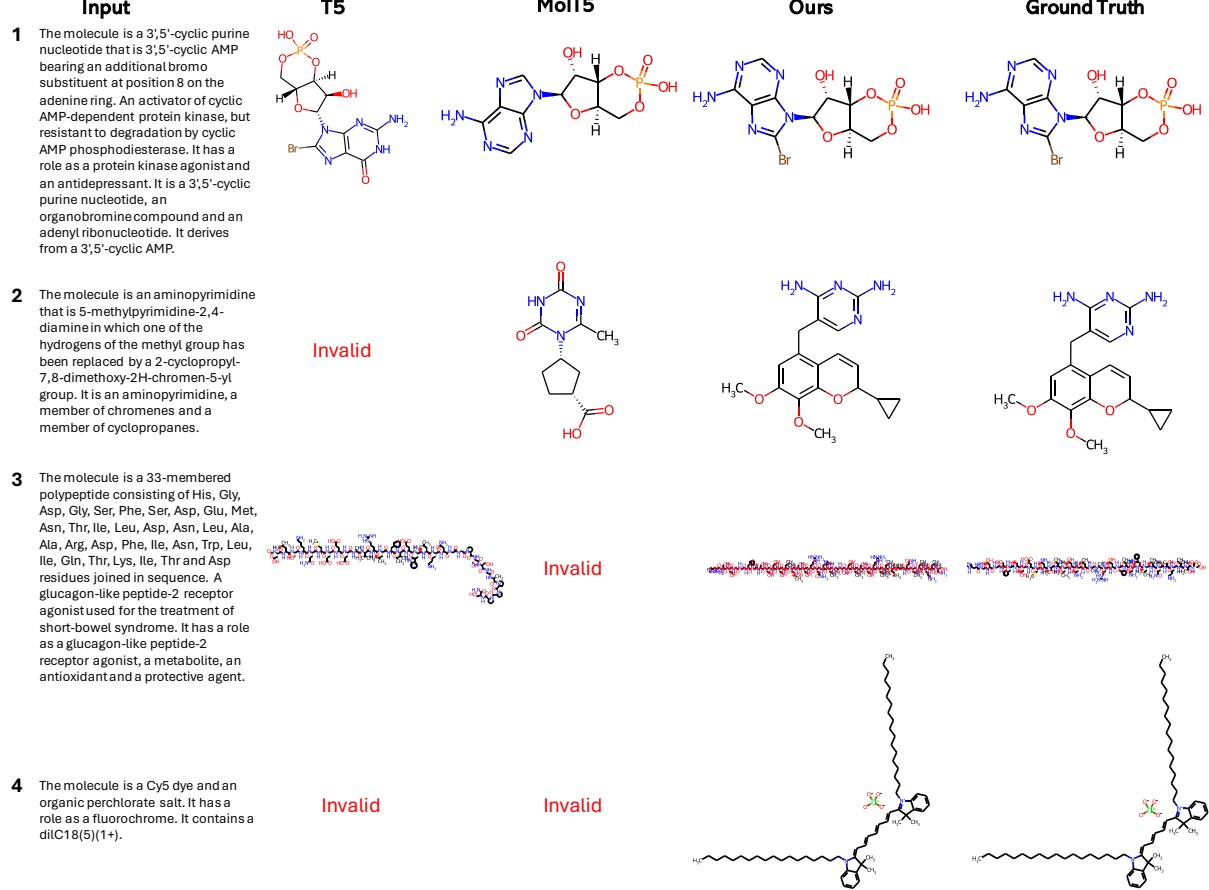

Figure 6: Text-based molecule generation cases.