# OpenReview forum: "BioT5: Enriching Cross-modal Integration in Biology with Chemical Knowledge and Natural Language Associations"
_EMNLP/2023/Conference — EMNLP 2023 Main_

### Official Review · Reviewer_tZxv · 2023-07-23

**Typos Grammar Style And Presentation Improvements:** 1. Is the title meant to "cross-model…
**Soundness:** 4

**Excitement:**

3: Ambivalent: It has merits (e.g., it reports state-of-the-art results, the idea is nice), but there are key weaknesses (e.g., it describes incremental work), and it can significantly benefit from another round of revision. However, I won't object to accepting it if my co-reviewers champion it.

**Paper Topic And Main Contributions:**

The paper develops a new model "BioT5" based on T5 for tasks involving  molecule / proteins combined with text. Finetuned BioT5 seems to achieve SOTA performance on most of the tasks and datasets listed in the paper.

**Questions For The Authors:**

A). Is the tokenisation ( separate vocabulary for general text. vs protein / chemical names) that is causing the boost in performance? or is it the set of  pretraining tasks ?

**Reasons To Accept:**

1. Finetuned BioT5 seems to achieve SOTA performance on most of the tasks and datasets listed in the paper.
2. The experiments and results are well documented.

**Reasons To Reject:**

1. Ablation study / analysis of what aspects are crucial to the performance gains would be helpful.

**Reproducibility:**

3: Could reproduce the results with some difficulty. The settings of parameters are underspecified or subjectively determined; the training/evaluation data are not widely available.

**Reviewer Confidence:**

2: Willing to defend my evaluation, but it is fairly likely that I missed some details, didn't understand some central points, or can't be sure about the novelty of the work.

---

> ### Author Rebuttal · Authors · 2023-08-28
>
> Thanks for your insightful comments. We appreciate your time and effort and would like to address your concerns.
>
> > **Ablation study / analysis of what aspects are crucial to the performance gains would be helpful. Q-A: Is the tokenisation ( separate vocabulary for general text. vs protein / chemical names) that is causing the boost in performance? or is it the set of pretraining tasks ?**
>
> Sorry for the lack of an ablation study, which is important to understand the effectiveness of each part of BioT5. Following your suggestion, we conduct the ablation study with the following 5 scenarios:
>
> - (a) Retaining only single-modal data, excluding wrapped text, molecule-text pairs, and protein-text pairs.
> - (b) Keeping single-modal data, molecule-text pairs, and protein-text pairs, excluding wrapped text.
> - (c) Keeping single-modal data and wrapped text, excluding molecule-text pairs and protein-text pairs.
> - (d) Shared vocabulary among three modalities: using the text tokenizer and vocabulary from original T5 [1] for molecule SELFIES tokens and protein amino acid tokens.
> - (e) Fine-tuning on downstream tasks from scratch.
>
> Then we fine-tune the above pre-trained models on molecule property prediction task with BACE and BBBP datasets and drug-target interaction (DTI) task with BioSNAP dataset. The results are shown in the following tables ("—" means removing the corresponding part).
>
> | Method/Dataset | BACE | BBBP |
> | --- | --- | --- |
> | BioT5 | 0.894 (0.003) | 0.777 (0.006) |
> | (a) —wrapped text —molecule text pairs —protein text pairs | 0.864 (0.008) | 0.740 (0.001) |
> | (b) —wrapped text | 0.887 (0.001) | 0.753 (0.009) |
> | (c) —molecule text pairs —protein text pairs | 0.876 (0.006) | 0.759 (0.013) |
> | (d) —separate chemical dictionary  | 0.862 (0.007) | 0.751 (0.007) |
> | (e) from scratch  | 0.836 （0.007） | 0.698 (0.001) |
>
> Table 1: Ablation study on molecule property prediction with BACE and BBBP dataset. The evaluation metric is AUROC.
>
> | Method | AUROC | AUPRC | Accuracy |
> | --- | --- | --- | --- |
> | BioT5 | 0.937 (0.001) | 0.937 (0.004) | 0.874 (0.004) |
> | (a) —wrapped text —molecule text pairs —protein text pairs | 0.920 (0.002) | 0.919 (0.003) | 0.855 (0.002) |
> | (b) —wrapped text | 0.928 (0.001) | 0.927 (0.002) | 0.864 (0.002) |
> | (c) —molecule text pairs —protein text pairs | 0.930 (0.002) | 0.929 (0.003) | 0.865 (0.005) |
> | (d) —separate chemical dictionary | 0.902 (0.009) | 0.908 (0.008) | 0.827 (0.015) |
> | (e) from scratch | 0.845 (0.004) | 0.862 (0.002) | 0.752 (0.018) |
>
> Table 2: Ablation study on drug-target interaction prediction with BioSNAP dataset.
>
> From the tables, we have the following observations.
>
> 1. **Importance of multimodal data**: The complete BioT5 model, which presumably encompasses all data modalities, outperforms all the ablated models. Scenario (a) demonstrates the significant drop in performance when removing wrapped text, molecule-text pairs, and protein-text pairs, thereby underscoring the value of these components in contributing to the model's performance.
> 2. **Role of wrapped text and molecule/protein-text pairs**: In scenario (b), when only wrapped text is excluded, there is a decline in performance relative to the full BioT5 model but is not as pronounced as when all the modalities are removed (as in scenario (a)). The same conclusion is also observed in scenario (c). This indicates that both wrapped text and molecule/protein-text pairs are important in capturing the properties and relationships of molecules and proteins
> 3. **Tokenization strategy**: Scenario (d) sheds light on the importance of the separate tokenization and dictionary for molecule SELFIES tokens, protein amino acid tokens, and text tokens. Utilizing the text tokenizer and vocabulary from the original T5 for bio-sequences yields lower performance, suggesting that our domain-specific tokenization strategy is beneficial for biological tasks.
> 4. **Value of pre-training**: The most drastic performance reduction is observed in scenario (e), where models are fine-tuned from scratch. This emphasizes the immense value of pre-training, even in domain-specific tasks.
>
> We appreciate your suggestion and will include the ablation study in the revised version.
>
> > **Typos**
> >
>
> Thank you for pointing out the potential typo and it should be “cross-modal”. We will revise it in the revised version of our manuscript.
>
> ********************References********************
>
> [1] Raffel, Colin, et al. "Exploring the limits of transfer learning with a unified text-to-text transformer." *The Journal of Machine Learning Research* 21.1 (2020): 5485-5551.

---

### Official Review · Reviewer_AqiU · 2023-08-01

**Soundness:** 4

**Excitement:**

4: Strong: This paper deepens the understanding of some phenomenon or lowers the barriers to an existing research direction.

**Paper Topic And Main Contributions:**

In this paper the authors propose BioT5, a T5 based model that integrates knowledge from chemical and protein textual representations, scientific literature, and from domain databases. The paper describes the model rationale, pre-training and fine-tuning settings and evaluates the model in several down-stream tasks. The paper is well organised and clear; previous works are presented in minimal detail; the proposed model and approach is described in sufficient detail.
The described pre-training framework results in very positive results, overcoming the compared works in most tasks, which demonstrates the ability of the model to integrate knowledge from the different sources.

**Questions For The Authors:**

A. What is the justification for choosing T5 as opposed to other architectures?
B. Other than "reduce the gap between pre-training and fine-tuning", is there a justification for adopting prompt-based fine-tuning? Would more task oriented fine-tuning approaches (e.g. standard classification setting)  be more well suited for some of the tasks?
C. Line 102: What does "text data from the biological domain" refers to? Is this referring to the scientific literature?


**Reasons To Accept:**

The methodology proposed achieves very positive results, which demonstrates the ability of the model to integrate knowledge from the different sources.
The use of separate tokenisation and embedding.
The work is clearly described and with good level of detail.

**Reasons To Reject:**

The use of prompt-based fine tuning, which is only lightly justified.
The lack of an ablation study, which can be understandable given the already extensive work and paper.

**Reproducibility:**

4: Could mostly reproduce the results, but there may be some variation because of sample variance or minor variations in their interpretation of the protocol or method.

**Reviewer Confidence:**

4: Quite sure. I tried to check the important points carefully. It's unlikely, though conceivable, that I missed something that should affect my ratings.

**Typos Grammar Style And Presentation Improvements:**

Typos:
line 264: sequneces
line 310: nature
line 549: those have

---

> ### Author Rebuttal · Authors · 2023-08-28
>
> We sincerely thank you for the positive feedback and valuable suggestions. Below we would like to give responses to your comments.
>
> > **The use of prompt-based fine tuning, which is only lightly justified.**
> **Q-B: Other than "reduce the gap between pre-training and fine-tuning", is there a justification for adopting prompt-based fine-tuning? Would more task oriented fine-tuning approaches (e.g. standard classification setting) be more well suited for some of the tasks?**
> >
>
> Thank you for this insightful question. Beyond the aim to reduce the gap between pre-training and fine-tuning, we have several reasons for adopting prompt-based fine-tuning:
>
> 1. **Simpleness and efficiency**: Compared to task-oriented fine-tuning approaches like standard classification settings, prompt-based fine-tuning doesn't require the introduction of additional layers, heads, or parameters. This makes the process more efficient and streamlined.
> 2. **Flexibility and extensibility**: Current GPT-based model training predominantly uses instruction tuning [3, 4, 5]. This approach is more extensible and can easily support various types of prompts, making it more user-friendly.
>
> > **The lack of an ablation study, which can be understandable given the already extensive work and paper.**
> >
>
> Thanks for this insightful comment. Following your suggestion, we conduct the ablation study with the following 5 scenarios:
>
> - (a) Retaining only single-modal data, excluding wrapped text, molecule-text pairs, and protein-text pairs.
> - (b) Keeping single-modal data, molecule-text pairs, and protein-text pairs, excluding wrapped text.
> - (c) Keeping single-modal data and wrapped text, excluding molecule-text pairs and protein-text pairs.
> - (d) Shared vocabulary among three modalities: using the text tokenizer and vocabulary from original T5 [1] for molecule SELFIES tokens and protein amino acid tokens.
> - (e) Fine-tuning on downstream tasks from scratch.
>
> Then we fine-tune the above pre-trained models on molecule property prediction task with BACE and BBBP datasets and drug-target interaction (DTI) task with BioSNAP dataset. The results are shown in the following tables ("—" means removing the corresponding part).
>
> | Method/Dataset | BACE | BBBP |
> | --- | --- | --- |
> | BioT5 | 0.894 (0.003) | 0.777 (0.006) |
> | (a) —wrapped text —molecule text pairs —protein text pairs | 0.864 (0.008) | 0.740 (0.001) |
> | (b) —wrapped text | 0.887 (0.001) | 0.753 (0.009) |
> | (c) —molecule text pairs —protein text pairs | 0.876 (0.006) | 0.759 (0.013) |
> | (d) —separate chemical dictionary  | 0.862 (0.007) | 0.751 (0.007) |
> | (e) from scratch  | 0.836 （0.007） | 0.698 (0.001) |
>
> Table 1: Ablation study on molecule property prediction with BACE and BBBP dataset. The evaluation metric is AUROC.
>
> | Method | AUROC | AUPRC | Accuracy |
> | --- | --- | --- | --- |
> | BioT5 | 0.937 (0.001) | 0.937 (0.004) | 0.874 (0.004) |
> | (a) —wrapped text —molecule text pairs —protein text pairs | 0.920 (0.002) | 0.919 (0.003) | 0.855 (0.002) |
> | (b) —wrapped text | 0.928 (0.001) | 0.927 (0.002) | 0.864 (0.002) |
> | (c) —molecule text pairs —protein text pairs | 0.930 (0.002) | 0.929 (0.003) | 0.865 (0.005) |
> | (d) —separate chemical dictionary | 0.902 (0.009) | 0.908 (0.008) | 0.827 (0.015) |
> | (e) from scratch | 0.845 (0.004) | 0.862 (0.002) | 0.752 (0.018) |
>
> Table 2: Ablation study on drug-target interaction prediction with BioSNAP dataset.
>
> From the tables, we have the following observations.
>
> 1. **Importance of multimodal data**: The complete BioT5 model, which presumably encompasses all data modalities, outperforms all the ablated models. Scenario (a) demonstrates the significant drop in performance when removing wrapped text, molecule-text pairs, and protein-text pairs, thereby underscoring the value of these components in contributing to the model's performance.
> 2. **Role of wrapped text and molecule/protein-text pairs**: In scenario (b), when only wrapped text is excluded, there is a decline in performance relative to the full BioT5 model but is not as pronounced as when all the modalities are removed (as in scenario (a)). The same conclusion is also observed in scenario (c). This indicates that both wrapped text and molecule/protein-text pairs are important in capturing the properties and relationships of molecules and proteins
> 3. **Tokenization strategy**: Scenario (d) sheds light on the importance of the separate tokenization and dictionary for molecule SELFIES tokens, protein amino acid tokens, and text tokens. Utilizing the text tokenizer and vocabulary from the original T5 for bio-sequences yields lower performance, suggesting that our domain-specific tokenization strategy is beneficial for biological tasks.
> 4. **Value of pre-training**: The most drastic performance reduction is observed in scenario (e), where models are fine-tuned from scratch. This emphasizes the immense value of pre-training, even in domain-specific tasks.
>
> We appreciate your suggestion and will include this ablation study in the revised version.
>
> > **Q-A: What is the justification for choosing T5 as opposed to other architectures?**
> >
>
> Thank you for raising this question. The choice of the T5 architecture for our BioT5 model was based on several considerations:
>
> 1. We don’t use encoder-only architecture since our downstream tasks include generative tasks such as molecule captioning and text-based molecule generation. This necessitated the use of an architecture that wasn't encoder-only.
> 2. Given that our BioT5 model (\~250M parameters) is not yet at the super scale of current large language models [6, 7], the bidirectional encoding capabilities of the T5 encoder are potentially beneficial. There is a comparison between encoder-decoder and decoder-only architectures in T5 [1] paper's Section 3.2 that encoder-decoder seems to be a better one at that scale (\~250M). We will explore the decoder-only model and scale up BioT5 in future work.
>
> > **Q-C: Line 102: What does "text data from the biological domain" refers to? Is this referring to the scientific literature?**
> >
>
> Yes. The "text data from the biological domain" refers to the biological articles/literature from PubMed [2]. PubMed is a biological database consisting of much literature on biomedical topics and life sciences. We will add a more detailed description in the revised version.
>
> > **Typos**
> >
>
> We appreciate your keen observation of the typos and will revise them in the later version of our manuscript.
>
> **References**
>
> [1] Raffel, Colin, et al. "Exploring the limits of transfer learning with a unified text-to-text transformer." *The Journal of Machine Learning Research* 21.1 (2020): 5485-5551.
>
> [2] Canese, Kathi, and Sarah Weis. "PubMed: the bibliographic database." *The NCBI handbook* 2.1 (2013).
>
> [3] Wei, Jason, et al. "Finetuned language models are zero-shot learners." *arXiv preprint arXiv:2109.01652* (2021).
>
> [4] Ouyang, Long, et al. "Training language models to follow instructions with human feedback." *Advances in Neural Information Processing Systems* 35 (2022): 27730-27744.
>
> [5] Peng, Baolin, et al. "Instruction tuning with gpt-4." arXiv preprint arXiv:2304.03277 (2023).
>
> [6] OpenAI. "GPT-4 Technical Report." arXiv preprint arXiv:2303.08774 (2023)
>
> [7] Touvron, Hugo, et al. "Llama 2: Open foundation and fine-tuned chat models." arXiv preprint arXiv:2307.09288 (2023).

---

### Official Review · Reviewer_fMEa · 2023-08-11

**Soundness:** 3

**Excitement:**

3: Ambivalent: It has merits (e.g., it reports state-of-the-art results, the idea is nice), but there are key weaknesses (e.g., it describes incremental work), and it can significantly benefit from another round of revision. However, I won't object to accepting it if my co-reviewers champion it.

**Missing References:**

For Drug-drug interactions comparison. https://academic.oup.com/bioinformatics/article/37/17/2651/6171181

**Paper Topic And Main Contributions:**


The paper proposes a transformer-based language model named BioT5, which integrates information about chemical and biological entities with natural language associations. The T5 model uses SELFIES for representing molecules, FASTA for proteins and text tokens for unstructured literature. The model is pretrained in a multi-task way on individual text modalities, text pairs and wrapped text. The authors claim that the language model outperforms baselines across a range of single-instance prediction, multi-instance prediction and cross-modal generation tasks when fine-tuned using a prompt-based fine tuning strategy.


**Questions For The Authors:**

1. What is the performance of the  ablation studies on multi-task pretraining?
2. What  is the  baseline with shared vocabularies and joint training?
3. The authors should provide  learning curves.
4. Is there any advantage of using "SELFIES" over "SMILES" representation of chemical in this work?
5. For MoleculeNet sets, the regression task is not considered. Why?



**Reasons To Accept:**

1. The introduction and related works sections are well-written, explaining the overall functional parts of the model and gaps in the existing literature.
2. The proposed method shows competitive performances compared to the existing baselines.
3. Multi-task pre-training to model the connections between the three modalities.
4. An appropriate tokenization method is used to represent the chemicals and protein sequences that maintain each modality's integrity.
5. Prompt-based finetuning strategy to unify all the downstream tasks.


**Reasons To Reject:**

1. The method proposed by the authors is interesting, but none of the parts used in the workflow is novel.

**Reproducibility:**

3: Could reproduce the results with some difficulty. The settings of parameters are underspecified or subjectively determined; the training/evaluation data are not widely available.

**Reviewer Confidence:**

4: Quite sure. I tried to check the important points carefully. It's unlikely, though conceivable, that I missed something that should affect my ratings.

---

> ### Author Rebuttal · Authors · 2023-08-28
>
> We sincerely thank you for the positive feedback and constructive suggestions. Below we give detailed responses to your comments.
>
> > **The method proposed by the authors is interesting, but none of the parts used in the workflow is novel.**
> >
>
> Thanks for raising this concern. We agree that some components are borrowed from existing ways. However, we would like to emphasize the following points:
>
> 1. **Unique combination and adaptation:** While individual components might have been explored in other contexts, the unique combination and adaptation of these components in the realm of bioinformatics is novel. Our BioT5, is not just a mere assembly of existing parts but a synergistic integration that achieves superior performance. Specifically, we have integrated both structured and unstructured data to enhance the interaction among different modalities. The large-scale unstructured wrapped text data from biological literature provides a wealth of knowledge about the properties of bio-entities and their interactions. Meanwhile, the structured sequence-text data derived from databases offers a comprehensive understanding of bio-entities. This integration, to the best of our knowledge, has not been previously explored in this manner, offering a fresh perspective on the problem.
> 2. **Separate tokenization of bio-sequences:** We have highlighted the issues other methods face when tokenizing bio-sequences (in paper Figure 3). Our additional ablation study (see below) underscores the importance of using specialized and separate tokenization and dictionaries for bio-sequences, providing valuable insights for future work.
> 3. **Robust molecular representation:** By adopting a 100% robust molecular representation, SELFIES, we ensure that the generated molecules are valid. This has been validated in downstream text-based molecule generation tasks.
>
> We hope these details can help clarify the innovations and insights of our work. Thank you for your feedback, and we will better illustrate these points in the revised version.
>
> > **Q1: What is the performance of the ablation studies on multi-task pre-training? Q2: What is the baseline with shared vocabularies and joint training?**
>
> We apologize for the oversight. Following your suggestion, we conduct the ablation study with the following 5 scenarios:
>
> - (a) Retaining only single-modal data, excluding wrapped text, molecule-text pairs, and protein-text pairs.
> - (b) Keeping single-modal data, molecule-text pairs, and protein-text pairs, excluding wrapped text.
> - (c) Keeping single-modal data and wrapped text, excluding molecule-text pairs and protein-text pairs.
> - (d) Shared vocabulary among three modalities: using the text tokenizer and vocabulary from original T5 [1] for molecule SELFIES tokens and protein amino acid tokens.
> - (e) Fine-tuning on downstream tasks from scratch.
>
> Then we fine-tune the above pre-trained models on molecule property prediction task with BACE and BBBP datasets and drug-target interaction (DTI) task with BioSNAP dataset. The results are shown in the following tables ("—" means removing the corresponding part).
>
> | Method/Dataset | BACE | BBBP |
> | --- | --- | --- |
> | BioT5 | 0.894 (0.003) | 0.777 (0.006) |
> | (a) —wrapped text —molecule text pairs —protein text pairs | 0.864 (0.008) | 0.740 (0.001) |
> | (b) —wrapped text | 0.887 (0.001) | 0.753 (0.009) |
> | (c) —molecule text pairs —protein text pairs | 0.876 (0.006) | 0.759 (0.013) |
> | (d) —separate chemical dictionary  | 0.862 (0.007) | 0.751 (0.007) |
> | (e) from scratch  | 0.836 （0.007） | 0.698 (0.001) |
>
> Table 1: Ablation study on molecule property prediction with BACE and BBBP dataset. The evaluation metric is AUROC.
>
> | Method | AUROC | AUPRC | Accuracy |
> | --- | --- | --- | --- |
> | BioT5 | 0.937 (0.001) | 0.937 (0.004) | 0.874 (0.004) |
> | (a) —wrapped text —molecule text pairs —protein text pairs | 0.920 (0.002) | 0.919 (0.003) | 0.855 (0.002) |
> | (b) —wrapped text | 0.928 (0.001) | 0.927 (0.002) | 0.864 (0.002) |
> | (c) —molecule text pairs —protein text pairs | 0.930 (0.002) | 0.929 (0.003) | 0.865 (0.005) |
> | (d) —separate chemical dictionary | 0.902 (0.009) | 0.908 (0.008) | 0.827 (0.015) |
> | (e) from scratch | 0.845 (0.004) | 0.862 (0.002) | 0.752 (0.018) |
>
> Table 2: Ablation study on drug-target interaction prediction with BioSNAP dataset.
>
> From the tables, we have the following observations.
>
> 1. **Importance of multimodal data**: The complete BioT5 model, which presumably encompasses all data modalities, outperforms all the ablated models. Scenario (a) demonstrates the significant drop in performance when removing wrapped text, molecule-text pairs, and protein-text pairs, thereby underscoring the value of these components in contributing to the model's performance.
> 2. **Role of wrapped text and molecule/protein-text pairs**: In scenario (b), when only wrapped text is excluded, there is a decline in performance relative to the full BioT5 model but is not as pronounced as when all the modalities are removed (as in scenario (a)). The same conclusion is also observed in scenario (c). This indicates that both wrapped text and molecule/protein-text pairs are important in capturing the properties and relationships of molecules and proteins
> 3. **Tokenization strategy**: Scenario (d) sheds light on the importance of the separate tokenization and dictionary for molecule SELFIES tokens, protein amino acid tokens, and text tokens. Utilizing the text tokenizer and vocabulary from the original T5 for bio-sequences yields lower performance, suggesting that our domain-specific tokenization strategy is beneficial for biological tasks.
> 4. **Value of pre-training**: The most drastic performance reduction is observed in scenario (e), where models are fine-tuned from scratch. This emphasizes the immense value of pre-training, even in domain-specific tasks.
>
> Thanks for your suggestions and we will include the ablation study in the revised version.
>
> > **Q3: The authors should provide learning curves.**
> >
>
> We appreciate your suggestion regarding the inclusion of learning curves. Due to the constraints of the rebuttal, we provide the learning curves for pre-training in tabular form below.
>
> | Step | Loss |
> | --- | --- |
> | 25k | 1.150 |
> | 50k | 0.828 |
> | 75k | 0.743 |
> | 100k | 0.732 |
> | 125k | 0.693 |
> | 150k | 0.694 |
> | 175k | 0.657 |
> | 200k | 0.676 |
> | 225k | 0.678 |
> | 250k | 0.652 |
> | 275k | 0.615 |
> | 300k | 0.579 |
> | 325k | 0.552 |
> | 350k | 0.541 |
>
> Table 3: Pre-training loss, including denoising loss and translation loss, which are jointly optimized. The loss is averaged every 25k steps.
>
> We can observe from the table that the overall pre-training loss exhibits a progressive and consistent decrease, indicating effective convergence of the model.
>
> We recognize the importance of visualizing the learning process and will include detailed learning curves for both pre-training and fine-tuning in the revised version of our manuscript.
>
> > **Q4: Is there any advantage of using "SELFIES" over "SMILES" representation of chemical in this work?**
> >
>
> Both SELFIES and SMILES are sequence representations of molecules. Their expressive power for molecules is comparable, as evidenced by experiments in ChemBERTa [2] on the molecule property prediction task. However, SELFIES has a distinct advantage in molecular generation tasks. Specifically, every valid SELFIES corresponds to a valid molecule. As shown in Table 6 of the paper, in the task of text-based molecule generation, by employing SELFIES, our model BioT5 achieved a 100% validity rate. In contrast, all baseline methods that utilized SMILES for molecular modeling often resulted in the generation of invalid molecules.
>
> > **Q5: For MoleculeNet sets, the regression task is not considered. Why?**
> >
>
> Regression tasks are inherently more challenging than classification tasks as they require predicting specific numerical values. While classification labels can be directly translated into natural language, regression tasks often necessitate specialized designs for encoding and tokenizing numerical values [3]. Our primary focus in this work is to validate the advantages of our multi-task pre-training framework, so we initially focused on classification downstream tasks that don't require additional design. We plan to address regression tasks in future work.
>
> > **Missing drug-drug interactions comparison**
> >
>
> Thank you for pointing out the missing comparison on drug-drug interactions (DDI). In the current version of our paper, we do not focus on the DDI downstream task based on the observation that the results for this task are already quite promising (with an AUROC of 0.9994 [4]). We hypothesize that such impressive results might not necessitate additional pre-training for enhancement. However, we acknowledge the importance of discussing and incorporating DDI-related work and results and will add them in the revised version.
>
> ********************References********************
>
> [1] Raffel, Colin, et al. "Exploring the limits of transfer learning with a unified text-to-text transformer." *The Journal of Machine Learning Research* 21.1 (2020): 5485-5551.
>
> [2] Chithrananda, Seyone, Gabriel Grand, and Bharath Ramsundar. "ChemBERTa: large-scale self-supervised pretraining for molecular property prediction." *arXiv preprint arXiv:2010.09885* (2020).
>
> [3] Born, Jannis, and Matteo Manica. "Regression Transformer enables concurrent sequence regression and generation for molecular language modelling." *Nature Machine Intelligence* 5.4 (2023): 432-444.
>
> [4] Chen, Yujie, et al. "MUFFIN: multi-scale feature fusion for drug–drug interaction prediction." *Bioinformatics* 37.17 (2021): 2651-2658.

---

### Official Review · Reviewer_BcEQ · 2023-08-12

**Soundness:** 4

**Excitement:**

4: Strong: This paper deepens the understanding of some phenomenon or lowers the barriers to an existing research direction.

**Paper Topic And Main Contributions:**

The paper presents a method to represent molecules, proteins and their associations from the text. The model is compared with various baselines on different datasets for various tasks and achieves state-of-the-art performance in some of them.

**Questions For The Authors:**

A: Do you use the official data splits of train, dev, test?

B: Do you use the original reported results of the baselines or the models are tested here, for in some cases the results are different, like MolTrans.

**Reasons To Accept:**

- The paper proposes a method to represent multiple modalities.

- The paper performs an extensive comparison with some the available methods on different datasets for various tasks.

**Reasons To Reject:**

- It is not clear where the improvement of results come from, from SELFIES, FASTA, or general text representations?

- I understand space is limited, but it would be useful to perform error analysis.

- More details are needed regarding how the baseline results are computed.

**Reproducibility:**

4: Could mostly reproduce the results, but there may be some variation because of sample variance or minor variations in their interpretation of the protocol or method.

**Reviewer Confidence:**

4: Quite sure. I tried to check the important points carefully. It's unlikely, though conceivable, that I missed something that should affect my ratings.

**Typos Grammar Style And Presentation Improvements:**

The dataset of molecule captioning (ChEBI-20) needs to be mentioned in the main text of article.

---

> ### Author Rebuttal · Authors · 2023-08-28
>
> We sincerely thank you for the valuable comments. Below we would like to provide some explanations to address your concerns.
>
> > **It is not clear where the improvement of results come from, from SELFIES, FASTA, or general text representations?**
>
> Our results primarily stem from several aspects, including not only the SELFIES, FASTA, and general text representations but also their implicit and explicit combinations. By applying the standard T5 [1] pre-training to molecule SELFIES, protein FASTA, and general text individually, we ensure the model's proficiency in each modality. Pre-training on wrapped biological text allows the model to implicitly learn the properties of bio-entities and capture potential connections between different bio-entities. Performing bi-directional translation pre-training tasks on structured molecule-text data and protein-text data makes BioT5 explicitly capture the connection between bio-entities and text. The tailored tokenization and dictionary for each modality ensure their integrity, particularly for molecule SELFIES and protein FASTA.
>
> To further demonstrate the effectiveness of each part of our BioT5, we also conduct an additional ablation study. Specifically, we focus on the following 5 scenarios:
>
> - (a) Retaining only single-modal data, excluding wrapped text, molecule-text pairs, and protein-text pairs.
> - (b) Keeping single-modal data, molecule-text pairs, and protein-text pairs, excluding wrapped text.
> - (c) Keeping single-modal data and wrapped text, excluding molecule-text pairs and protein-text pairs.
> - (d) Shared vocabulary among three modalities: using the text tokenizer and vocabulary from original T5 [1] for molecule SELFIES tokens and protein amino acid tokens.
> - (e) Fine-tuning on downstream tasks from scratch.
>
> Then we fine-tune the above pre-trained models on molecule property prediction task with BACE and BBBP datasets and drug-target interaction (DTI) task with BioSNAP dataset. The results are shown in the following tables ("—" means removing the corresponding part).
>
> | Method/Dataset | BACE | BBBP |
> | --- | --- | --- |
> | BioT5 | 0.894 (0.003) | 0.777 (0.006) |
> | (a) —wrapped text —molecule text pairs —protein text pairs | 0.864 (0.008) | 0.740 (0.001) |
> | (b) —wrapped text | 0.887 (0.001) | 0.753 (0.009) |
> | (c) —molecule text pairs —protein text pairs | 0.876 (0.006) | 0.759 (0.013) |
> | (d) —separate chemical dictionary  | 0.862 (0.007) | 0.751 (0.007) |
> | (e) from scratch  | 0.836 （0.007） | 0.698 (0.001) |
>
> Table 1: Ablation study on molecule property prediction with BACE and BBBP dataset. The evaluation metric is AUROC.
>
> | Method | AUROC | AUPRC | Accuracy |
> | --- | --- | --- | --- |
> | BioT5 | 0.937 (0.001) | 0.937 (0.004) | 0.874 (0.004) |
> | (a) —wrapped text —molecule text pairs —protein text pairs | 0.920 (0.002) | 0.919 (0.003) | 0.855 (0.002) |
> | (b) —wrapped text | 0.928 (0.001) | 0.927 (0.002) | 0.864 (0.002) |
> | (c) —molecule text pairs —protein text pairs | 0.930 (0.002) | 0.929 (0.003) | 0.865 (0.005) |
> | (d) —separate chemical dictionary | 0.902 (0.009) | 0.908 (0.008) | 0.827 (0.015) |
> | (e) from scratch | 0.845 (0.004) | 0.862 (0.002) | 0.752 (0.018) |
>
> Table 2: Ablation study on drug-target interaction prediction with BioSNAP dataset.
>
> From the tables, we have the following observations.
>
> 1. **Importance of multimodal data**: The complete BioT5 model, which presumably encompasses all data modalities, outperforms all the ablated models. Scenario (a) demonstrates the significant drop in performance when removing wrapped text, molecule-text pairs, and protein-text pairs, thereby underscoring the value of these components in contributing to the model's performance.
> 2. **Role of wrapped text and molecule/protein-text pairs**: In scenario (b), when only wrapped text is excluded, there is a decline in performance relative to the full BioT5 model but is not as pronounced as when all the modalities are removed (as in scenario (a)). The same conclusion is also observed in scenario (c). This indicates that both wrapped text and molecule/protein-text pairs are important in capturing the properties and relationships of molecules and proteins
> 3. **Tokenization strategy**: Scenario (d) sheds light on the importance of the separate tokenization and dictionary for molecule SELFIES tokens, protein amino acid tokens, and text tokens. Utilizing the text tokenizer and vocabulary from the original T5 for bio-sequences yields lower performance, suggesting that our domain-specific tokenization strategy is beneficial for biological tasks.
> 4. **Value of pre-training**: The most drastic performance reduction is observed in scenario (e), where models are fine-tuned from scratch. This emphasizes the immense value of pre-training, even in domain-specific tasks.
>
> We hope the above explanation and ablation study can address your question and will add these in the revised version.
>
> > **I understand space is limited, but it would be useful to perform error analysis.**
> >
>
> We acknowledge the importance of error analysis. Here we conduct an analysis of drug-target interaction task with Human dataset. Following the DrugBAN [5], we fine-tune BioT5 on Human dataset with 5 random seeds and compare our results with the baseline method DrugBAN (the results of its five random runs are derived in its supplementary material).
>
> | Methods/Random Experiments | 1 | 2 | 3 | 4 | 5 | Mean±Std | p-value of t-test between BioT5 and DrugBAN |
> | --- | --- | --- | --- | --- | --- | --- | --- |
> | AUROC (BioT5) | 0.9899 | 0.9872 | 0.9886 | 0.9890 | 0.9882 | 0.989±0.001 | 0.0007 |
> | AUROC (DrugBAN) | 0.9836 | 0.9801 | 0.9849 | 0.9832 | 0.9790 | 0.982±0.002 |  |
> | AUPRC (BioT5) | 0.9873 | 0.9849 | 0.9833 | 0.9873 | 0.9843 | 0.985±0.002 | 0.0143 |
> | AUPRC (DrugBAN) | 0.9796 | 0.9762 | 0.9842 | 0.9828 | 0.9780 | 0.980±0.003 |  |
>
> From the table, we can observe the following outcomes:
>
> 1. For the AUROC, BioT5 achieved an AUROC score of 0.989±0.001, while DrugBAN attained an AUROC of 0.982±0.002. The t-test comparing the two results yielded a highly significant p-value of 0.0007.
> 2. For the AUPRC, BioT5 demonstrated an AUPRC score of 0.985±0.002. In contrast, DrugBAN recorded an AUPRC of 0.980±0.003. A subsequent t-test between these scores revealed a p-value of 0.0143.
>
> Both the AUROC and AUPRC results show statistically significant differences between BioT5 and DrugBAN, as indicated by their respective p-values being below the 0.05 threshold.
> Based on the observed results, BioT5 outperforms DrugBAN in both AUROC and AUPRC metrics, indicating superior classification performance and precision-recall trade-off for BioT5.
>
> We will incorporate more error analyses in the revised version of our manuscript.
>
> > **More details are needed regarding how the baseline results are computed. Q-B: Do you use the original reported results of the baselines or the models are tested here, for in some cases the results are different, like MolTrans.**
>
>
> Apologies for the oversight. Our results primarily derive from the original paper and use the same split method as the compared benchmark. Specifically,
>
> - For molecule property prediction experiments, we follow the commonly used scaffold split for MoleculeNet [2] benchmark, with results mainly sourced from MolXPT [3].
> - For protein property prediction and protein-protein interaction experiments, we adhere to the PEER [4] benchmark's split setting, with results also primarily from PEER.
> - For drug-target interaction experiments, our split method and results mainly follow DrugBAN [5]. For instance, the MolTrans [6] results are from the DrugBAN paper, with discrepancies arising from DrugBAN's reproduction of these baselines.
> - For molecule captioning and text-based molecule generation experiments, we follow the default split setting of the ChEBI-20 dataset (first proposed by Text2Mol [9]), with results primarily from MolT5 [7], MolXPT [3], and MolReGPT [8].
>
> We will provide a detailed description in the revised version of our manuscript.
>
> > **Q-A: Do you use the official data splits of train, dev, test?**
> >
>
> Yes, we use the same official split method as the compared benchmark.
>
> - For molecule property prediction experiments, we use the official DeepChem python package from [2] to get data splits.
> - For protein property prediction and protein-protein interaction experiments, we adhere to the PEER [4] benchmark's split setting as described in Table 1 of [4].
> - For drug-target interaction experiments, we follow DrugBAN [5] as described in Section “Evaluation strategies and metrics” of [5].
> - For molecule captioning and text-based molecule generation experiments, we use the default split setting of the CheBI-20 dataset. This dataset was first proposed by Text2Mol [9] and the default splitting was also used in MolT5 [7], MolXPT [3], etc.
>
> We will provide a detailed description in the revised version of our manuscript.
>
> > **The dataset of molecule captioning (ChEBI-20) needs to be mentioned in the main text of article**
>
> Thank you for pointing out the oversight regarding the dataset of molecule captioning (ChEBI-20). Due to space limitations, we briefly mentioned it at the beginning of Section 4.3. We will ensure to provide a more detailed description in the revised version of our manuscript.
>
> ********************References********************
>
> [1] Raffel, Colin, et al. "Exploring the limits of transfer learning with a unified text-to-text transformer." *The Journal of Machine Learning Research* 21.1 (2020): 5485-5551.
>
> [2] Wu, Zhenqin, et al. "MoleculeNet: a benchmark for molecular machine learning." *Chemical science* 9.2 (2018): 513-530.
>
> [3] Liu, Zequn, et al. "MolXPT: Wrapping Molecules with Text for Generative Pre-training." *arXiv preprint arXiv:2305.10688* (2023).
>
> [4] Xu, Minghao, et al. "Peer: a comprehensive and multi-task benchmark for protein sequence understanding." *Advances in Neural Information Processing Systems* 35 (2022): 35156-35173.
>
> [5] Bai, Peizhen, et al. "Interpretable bilinear attention network with domain adaptation improves drug–target prediction." *Nature Machine Intelligence* 5.2 (2023): 126-136.
>
> [6] Huang, Kexin, et al. "MolTrans: molecular interaction transformer for drug–target interaction prediction." *Bioinformatics* 37.6 (2021): 830-836.
>
> [7] Edwards, Carl, et al. "Translation between molecules and natural language." *arXiv preprint arXiv:2204.11817* (2022).
>
> [8] Li, Jiatong, et al. "Empowering Molecule Discovery for Molecule-Caption Translation with Large Language Models: A ChatGPT Perspective." *arXiv preprint arXiv:2306.06615* (2023).
>
> [9] Edwards, Carl, ChengXiang Zhai, and Heng Ji. "Text2mol: Cross-modal molecule retrieval with natural language queries." *Proceedings of the 2021 Conference on Empirical Methods in Natural Language Processing*. 2021.

---

### Meta-Review · Area_Chair_5ZCr · 2023-09-06

**Recommendation:** 4

**Metareview:**

The paper proposed a method to represent multiple modalities, perform an extensive comparison with existing methods, and demonstrate competitive performance. However, this paper has limitations in a lack of clarity on the source of result improvements, the absence of error analysis, the need for more details on baseline results computation, and a perceived lack of novelty in the components used in the workflow.

---

### Decision · Program_Chairs · 2023-10-07

**Decision:**

Accept-Main

**Comment:**

The paper proposed a method to represent multiple modalities, perform an extensive comparison with existing methods, and demonstrate competitive performance. However, this paper has limitations in a lack of clarity on the source of result improvements, the absence of error analysis, the need for more details on baseline results computation, and a perceived lack of novelty in the components used in the workflow.